# Efficient Fine-Grained Sampling Guidance for Diffusion-Based Symbolic Music Generation

## Abstract

The problem of symbolic music generation presents unique challenges due to the combination of limited data availability and the need for high precision in note pitch. To address these issues, we introduce an efficient Fine-grained Sampling Guidance (FTG) approach within diffusion models. FTG guides the diffusion models to generate music that aligns more closely with the control and intent of human composers, thereby improving the accuracy and quality of music generation. This method empowers diffusion models to excel in advanced applications such as improvisation, and interactive music creation. We derive theoretical characterizations for both the challenges in symbolic music generation and the effect of the FTG approach. We provide numerical experiments, subjective evaluation and a demo page [1] for interactive music generation with user input to showcase the effectiveness of our approach.

## 1 Introduction

Symbolic music generation is a subfield of music generation that focuses on creating music in symbolic form, typically represented as sequences of discrete events such as notes, pitches, rhythms, and durations. These representations are analogous to traditional sheet music or MIDI files, where the structure of the music is defined by explicit musical symbols rather than audio waveforms. Symbolic music generation has a wide range of applications, including automatic composition, music accompaniment and improvisation. It can also play a significant role in interactive music systems, where a model can respond to user inputs or generate improvisational passages in real-time. A lot of progress has been made in the field of deep symbolic music generation in recent years; see Huang et al. (2018), Min et al. (2023), von Rütte et al. (2023), Wang et al. (2024) and Huang et al. (2024).

Despite recent progress, some unique challenges of symbolic music generation remain unresolved. A key obstacle is the scarcity of high-quality training data. While large audio datasets are readily available, symbolic music data is more limited, often due to copyright constraints and the effort needed to create data. Additionally, unlike image generation, where the inaccuracy of a single pixel may not significantly affect overall quality, symbolic music generation demands high precision, especially in terms of pitch. In many tonal contexts, a single incorrect note can be glaringly obvious, even to less-trained ears.

As a partial motivation, we empirically observe the occurrence of "wrong notes" in existing state-of-the-art symbolic music generation models. We provide theoretical explanations for why these models often fail to avoid such errors. Apart from that, we find that many models encounter challenges in generating well-regularized accompaniment. While human-composed accompaniment often exhibits consistent patterns across bars and phrases, the generated symbolic accompaniment tends to vary significantly. These observations and theoretical discoveries motivate the method of applying regularization through external guidance, rather than relying on the model to capture it entirely autonomously.

We then address the precision challenge in symbolic music generation building upon a diffusion model-based approach. Diffusion models can flexibly capture a wide variety of patterns in the data distribution, and therefore generate highly structured and detailed images (Ho et al., 2020). This flexibility makes diffusion models well-suited for piano roll-based symbolic music generation, where

---

[1] https://huggingface.co/spaces/interactive-symbolic-music/InteractiveSymbolicMusicDemo

segmented piano rolls can be treated similarly to image data for processing. Further, guidance can be incorporated into the training process as background information and into the gradual denoising process to direct the sampling (Zhang et al., 2023b), enabling the design of specialized structures within diffusion models that integrate harmonic and rhythmic guidance. Our results in this work are summarized as follows:

- **Motivation**: We provide empirical observations and statistical theory evidence to reveal and characterize the precision and regularization challenges in symbolic music generation, underscoring the value of fine-grained guidance in training and generation.

- **Methodology**: We propose a controlled diffusion model for symbolic music generation that incorporates fine-grained harmonic and rhythmic guidance and regularization, in both the training and sampling processes. With limited training data, the model is capable of generating music with high accuracy, consistent rhythmic patterns, and even out-of-sample styles that align closely with the user's intent.

- **Effectiveness**: We provide both theoretical and empirical evidence supporting the effectiveness of our approach, and further demonstrate the potential of the model to be applied in interactive music systems, where the model efficiently and reliably integrates user-designed controls and generates improvisational passages in real-time.

### 1.1 RELATED WORK

**Symbolic music generation.** Symbolic music generation literature can be classified based on the choice of data representation, among which the MIDI token-based representation adopts a sequential discrete data structure, and is often combined with sequential generative models such as Transformers and LSTMs. Examples of works using MIDI token-based data representation include Huang et al. (2018), Huang & Yang (2020), Ren et al. (2020), Choi et al. (2020), Hsiao et al. (2021), Lv et al. (2023) and von Rütte et al. (2023). While the MIDI token-based representation enables generative flexibility, it also introduces the challenge of simultaneously learning multiple dimensions that exhibit significant heterogeneity, such as the "pitch" dimension compared to the "duration" dimension. An alternative data representation used in music processing is the piano roll-based format. Many recent works adopt this data representation; see Min et al. (2023), Zhang et al. (2023a), Wang et al. (2024) and Huang et al. (2024) for example. Our work differs from their works in that we apply the textural guidance jointly in both the training and sampling process, and with an emphasis on enhancing real-time generation precision and speed. More detailed comparisons are provided in Appendix D, after we present a comprehensive description of our methodology.

**Controlled diffusion models.** Multiple works in controlled diffusion models are related to our work in terms of methodology. Specifically, we adopt the idea of classifier-free guidance in training and generation, see Ho & Salimans (2022). To control the sampling process, Chung et al. (2022), Song et al. (2023) and Novack et al. (2024) guide the intermediate sampling steps using the gradients of a loss function. In contrast, Dhariwal & Nichol (2021), Saharia et al. (2022), Lou & Ermon (2023) and Fishman et al. (2023) apply projection and reflection during the sampling process to straightforwardly incorporate data constraints. Different from these works, we design guidance for intermediate steps tailored to the unique characteristics of symbolic music data and generation. While the meaning of a specific pixel in an image is undefined until the entire image is generated, each position on a piano roll corresponds to a fixed time-pitch pair from the outset. This new context enables us to develop novel implementations and theoretical perspectives on the guidance approach.

## 2 BACKGROUND: DIFFUSION MODELS FOR PIANO ROLL GENERATION

In this section, we introduce the data representation of piano roll. We then introduce the formulations of diffusion model, combined with an application on modeling the piano roll data.

Let $\mathbf{M} \in \{0, 1\}^{L \times H}$ be a piano roll segment, where $H$ is the pitch range and $L$ is the number of time units in a frame. For example, $H$ can be set as $128$, representing a pitch range of $0 - 127$, and $L$ can be set as $64$, representing a 4-bar segment with time signature 4/4 (4 beats per bar) and 16th-note resolution. Each element $\mathbf{M}_{lh}$ of $\mathbf{M}$ ($1 \le l \le L$, $1 \le h \le H$) takes value 0

or 1, where $M_{lh} = 1/0$ represents the presence/absence of a note at time index $l$ and pitch $h^2$. Since standard diffusion models are based on Gaussian noise, the output of the diffusion model is a continuous random matrix $\mathbf{X} \in \mathbb{R}^{L \times H}$, which is then projected to the discrete piano roll $\mathbf{M}$ by $M_{lh}(\mathbf{X}) = \mathbf{1}\{X_{lh} \geq 1/2\}$, where $\mathbf{1}\{\cdot\}$ stands for the indicator function.

To model and generate the distribution of $\mathbf{M}$, denoted as $P_{\mathbf{M}}$, we use the the Denoising Diffusion Probabilistic Modeling (DDPM) formulation (Ho et al., 2020). The objective of DDPM training, with specific choices of parameters and reparameterizations, is given as

$$\mathbb{E}_{t \sim \mathcal{U}[\![1,T]\!], \mathbf{X}_0 \sim P_{\mathbf{M}}, \boldsymbol{\varepsilon} \sim \mathcal{N}(0, \mathbf{I})}[\lambda(t)\|\boldsymbol{\varepsilon} - \boldsymbol{\varepsilon}_\theta(\mathbf{X}_t, t)\|^2], \qquad (1)$$

where $\mathbf{X}_t = \sqrt{\bar{\alpha}_t}\mathbf{X}_0 + \sqrt{1 - \bar{\alpha}_t}\boldsymbol{\varepsilon}$ with hyperparameters $\{\beta_t\}$, $\bar{\alpha}_t = \prod_{s=0}^{t}(1 - \beta_s)$, and $\boldsymbol{\varepsilon}_\theta$ is a deep neural network with parameter $\theta$. Moreover, according to the connection between diffusion models and score matching (Song & Ermon, 2019), the deep neural network $\boldsymbol{\varepsilon}_\theta$ can be used to derive an estimator of the score function $\boldsymbol{s}_t(\mathbf{X}_t) = \nabla_{\mathbf{X}_t} \log p_t(\mathbf{X}_t)$. Specifically, $\boldsymbol{s}_t(\mathbf{X}_t)$ can be approximated by $-\boldsymbol{\varepsilon}_\theta(\mathbf{X}_t, t)/\sqrt{1 - \bar{\alpha}_t}$.

With the trained noise prediction network $\boldsymbol{\varepsilon}_\theta$, the reverse sampling process can be formulated as (Song et al., 2020a):

$$\mathbf{X}_{t-1} = \sqrt{\bar{\alpha}_{t-1}}\left(\frac{\mathbf{X}_t - \sqrt{1 - \bar{\alpha}_t}\boldsymbol{\varepsilon}_\theta(\mathbf{X}_t, t)}{\sqrt{\bar{\alpha}_t}}\right) + \sqrt{1 - \bar{\alpha}_{t-1} - \sigma_t^2}\boldsymbol{\varepsilon}_\theta(\mathbf{X}_t, t) + \sigma_t\boldsymbol{\varepsilon}_t, \qquad (2)$$

where $\sigma_t$ are hyperparameters chosen corresponding to equation 1, and $\boldsymbol{\varepsilon}_t$ is standard Gaussian noise at each step. Going backward in time from $\mathbf{X}_T \sim \mathcal{N}(0, \mathbf{I})$, the process yields the final output $\mathbf{X}_0$, which can be converted into a piano roll $\mathbf{M}(\mathbf{X}_0)$.

According to Song et al. (2020b), the DDPM forward and backward processes can be regarded as discretizations of the following SDEs:

$$d\mathbf{X}_t = -\frac{1}{2}\beta(t)\mathbf{X}_t dt + \sqrt{\beta(t)}d\mathbf{W}_t, \qquad (3)$$

$$d\mathbf{X}_t = -\left[\frac{1}{2}\beta(t)\mathbf{X}_t + \beta(t)\boldsymbol{s}_t(\mathbf{X}_t)\right]dt + \sqrt{\beta(t)}d\bar{\mathbf{W}}_t, \qquad (4)$$

## 3 CHALLENGES IN SYMBOLIC MUSIC GENERATION

While generative models have achieved significant success in text, image, and audio generation, the effective modeling and generation of symbolic music remains a relatively unexplored area. In this section, we introduce two major challenges of current symbolic music generation.

### 3.1 HARMONIC PRECISION

One challenge of symbolic music generation involves the high precision required for music generation. Specifically, harmony considerations serve as an illustrative example for highlighting the issue with precision. In music, harmony refers to the simultaneous sound of different notes that form a cohesive entity in the mind of the listener (Müller, 2015). Unlike image generation, where a slightly misplaced pixel may not significantly affect the overall image quality, an "inaccurately" generated musical note can drastically disrupt the harmony, affecting the quality of a piece.

To demonstrate the issue in harmonic precision, we consider temporary tonic key signatures[3], which establish the tonal center of music. In many genres[4], out-of-key notes are uncommon, and produce noticeable dissonance without a suitable context. For instance, a G♮ note is considered as out-of-key

---

[2] This is a slightly simplified representation model for the purpose of theoretical analysis, the specified version with implementation details is provided in Section 5.1

[3] As a clarification, instead of assigning one single key to a piece or a big section, here we refer to each key associated with the *temporary tonic*.

[4] We note that out-of-key notes are more common in genres such as jazz and contemporary music. However, symbolic datasets rarely include music from these genres. Further, their inherent flexibility and the ambiguity in the assessment of quality present additional challenges for generative models. As a result, these genres are beyond the scope of this work.

in a G♭ major context. While such notes might add an interesting tonal color when intentionally used by composers, they are usually perceived merely as mistakes when appearing in generative model outputs, as demonstrated by some examples on our demo page.

**Why generative models struggle with out-of-key notes**   In this section, we characterize why an out-of-key note is unlikely to be generated in a way that sounds "right" in context by a symbolic music generation model. We note that a summary of non-standard notations is provided in Appendix A. Denote the probability of $\mathbf{M} = M$ as $P_{\mathbf{M}}(M)$. Let $P_{\mathbf{M}}(\boldsymbol{w})$ denote the probability that $M$ has at least one note-out-of-key. The inclusion of a note-out-of-key requires a meticulously crafted surrounding context in order to function as a legitimate accidental, rather than being perceived as a mere error. Let $P_{\mathbf{M}}(\boldsymbol{w}, \boldsymbol{c})$ denote the probability that there is a surrounding context accommodating the existence of out-of-key notes (referred to in brief as *"accommodating context"*). We now consider the probability of not having an "accommodating context", given that out-of-key notes are generated, i.e., $P_{\mathbf{M}}(\bar{\boldsymbol{c}}|\boldsymbol{w})$. In this case, the out-of-key notes are likely perceived as "wrong notes", due to the lack of an accommodating context. Denote the estimated distributions and probabilistic values with $\widehat{P}_M(\cdot)$, we have

$$\widehat{P}_{\mathbf{M}}(\bar{\boldsymbol{c}}|\boldsymbol{w}) = \frac{\widehat{P}_{\mathbf{M}}(\bar{\boldsymbol{c}}, \boldsymbol{w})}{\widehat{P}_{\mathbf{M}}(\boldsymbol{w})} = \frac{\widehat{P}_{\mathbf{M}}(\bar{\boldsymbol{c}}, \boldsymbol{w})}{\widehat{P}_{\mathbf{M}}(\boldsymbol{c}, \boldsymbol{w}) + \widehat{P}_{\mathbf{M}}(\bar{\boldsymbol{c}}, \boldsymbol{w})}.$$

In practice, $\widehat{P}_{\mathbf{M}}(\boldsymbol{c}, \boldsymbol{w})$ is very small, as an accommodating context requires the careful design and precise generation of each pixel on the $L \times H$ canvas. Therefore, when the modeling error in $\widehat{P}_{\mathbf{M}}(\bar{\boldsymbol{c}}, \boldsymbol{w})$ is large, $\widehat{P}_{\mathbf{M}}(\bar{\boldsymbol{c}}|\boldsymbol{w})$ is close to 1, meaning almost every out-of-key note generated by the model is likely perceived as a "wrong note". The following proposition 1 further provides theoretical characterization of the lower-bound of $\widehat{P}_{\mathbf{M}}(\bar{\boldsymbol{c}}, \boldsymbol{w})$, where $n^{-1/(LH+2)}$ implies slow decrease of estimation error (in general $LH = 128 \times 128$). The probability class $\mathcal{P}_\delta$ is the search space of a continuous model[5] $\widehat{p}_{\mathbf{X}}$. The proof and details of $\mathcal{P}_\delta$ are given in appendix B.1

**Proposition 1.** *Consider generating $P_{\mathbf{M}}$ from a continuous random variable $\mathbf{X}$, i.e., given $n$ i.i.d. data $\{\mathbf{X}^i\}_{i=1}^n \sim p_{\mathbf{X}}$, let $\{\mathbf{M}^i\}_{i=1}^n$ be given by $\mathbf{M}_{lh}^i = \mathbf{1}\{\mathbf{X}_{lh}^i \geq 1/2\}$. Denote the model for estimating the distribution of $\mathbf{X}$ as $\widehat{p}_{\mathbf{X}}$. We have $\exists C > 0$ such that $\forall n$,*

$$\inf_{\widehat{p}_{\mathbf{X}}} \sup_{p_{\mathbf{X}} \in \mathcal{P}_\delta} \mathbb{E}_{\{\mathbf{M}^i\}_{i=1}^n \sim P_{\mathbf{M}}} \widehat{P}_{\mathbf{M}}(\bar{\boldsymbol{c}}, \boldsymbol{w}) \geq C \cdot n^{-\frac{1}{LH+2}} - P_{\mathbf{M}}(\bar{\boldsymbol{c}}, \boldsymbol{w}), \tag{5}$$

*where $\widehat{P}_{\mathbf{M}}$ is derived from $\widehat{p}_{\mathbf{X}}$ via the connection $\widehat{\mathbf{M}}_{lh}^i = \mathbf{1}\{\widehat{\mathbf{X}}_{lh}^i \geq 1/2\}$.*

To empirically support our analysis, we provide samples on our demo page that, despite generated by well-trained and well-conditioned diffusion models, still include out-of-key notes likely perceived by human listeners as dissonant mistakes.

## 3.2 Rhythmic regularity

A second observation regarding symbolic music generation models is their tendency to produce irregular rhythmic patterns. While many composers typically maintain consistent rhythmic patterns across consecutive measures within a 4-bar phrase, particularly in the accompaniment, such variations frequently appear in the generated accompaniment of symbolic music generative models.

Such phenomenons can be explained by the scarcity of data and the high dimensionality hindering the model's ability to capture correlations between different bars, even within a single generated section. Additionally, the irregularity in generated patterns can stem from the presence of irregular samples in many existing MIDI datasets. Without a sufficient quantity of data exhibiting clear correlations and repetition across measures, it is challenging for the model to self-generate more human-like and consistent accompaniment patterns.

---

[5]While music data is discrete, many existing works (Wang et al., 2024; Huang et al., 2024) suggest that continuous diffusion models outperform discrete diffusion models (Lv et al., 2023) in symbolic music generation. This justifies modeling the search space of the model with a class of continuous probability distributions.

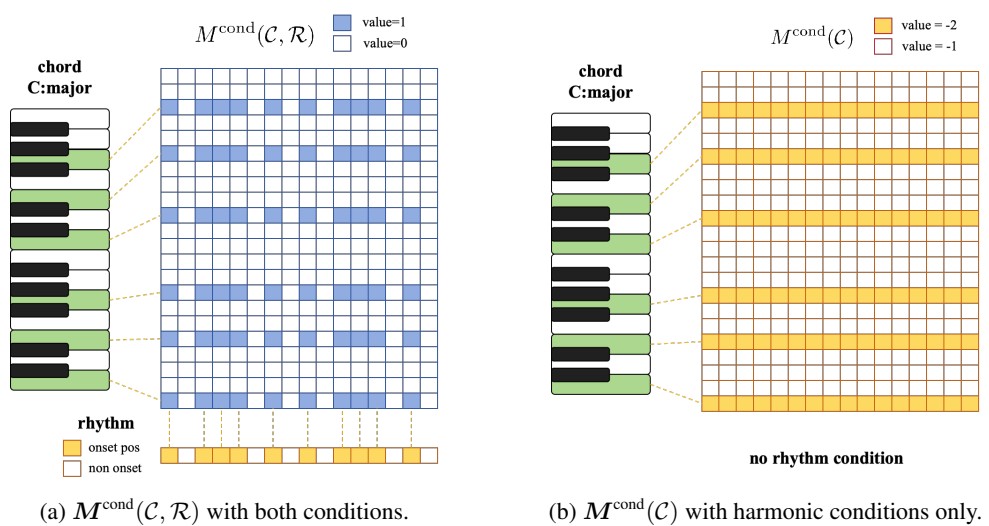

Figure 1: Example of an accompaniment segment generated by a diffusion model depicting high variation in rhythmic pattern.

## 4 METHODOLOGY: FINE-GRAINED TEXTURAL GUIDANCE

In the previous section 3, we identified the unique challenges in symbolic music generation arising from the distinctive characteristics and specific requirements of symbolic music data. Together with the scarcity of available high-quality data for training, this creates a motivation for fine-grained external control and regularization in generating symbolic music. In this section, we present our methodology of applying fine-grained regularization guidance to improve the quality and stability of the generated symbolic music, ensuring better alignment with the user's intent. Specifically, we design fine-grained conditioning and sampling correction/regularization, altogether referred to as *Fine-grained Textural Guidance* (FTG) that leverage this characteristic of the piano roll data. We use "texture" to refer to harmony and rhythm together.

### 4.1 FINE-GRAINED CONDITIONING IN TRAINING

We train a conditional diffusion model with fine-grained harmonic ($\mathcal{C}$, required) and rhythmic ($\mathcal{R}$, optional) conditions, which are provided to the diffusion models in the form of a piano roll $M^{\mathrm{cond}}$. We provide illustration of $M^{\mathrm{cond}}(\mathcal{C}, \mathcal{R})$ and $M^{\mathrm{cond}}(\mathcal{C})$ via examples if Figure 2. The mathematical descriptions are provided in Appendix C.

(a) $M^{\mathrm{cond}}(\mathcal{C}, \mathcal{R})$ with both conditions.  (b) $M^{\mathrm{cond}}(\mathcal{C})$ with harmonic conditions only.

Figure 2: An illustrative example of $M^{\mathrm{cond}}(\mathcal{C}, \mathcal{R})$ and $M^{\mathrm{cond}}(\mathcal{C})$.

Moreover, to enable the model to generate under varying levels of conditioning, including unconditional generation, we implement the idea of classifier-free guidance, and randomly apply conditions with or without rhythmic pattern in the process of training. Namely, the training loss is modified from equation 1 and given as

$$\mathbb{E}_{t, \boldsymbol{\varepsilon}, \mathbf{X}_0}[\lambda_1(t)\|\boldsymbol{\varepsilon} - \boldsymbol{\varepsilon}_\theta(\mathbf{X}_t, \mathbf{M}^{\mathrm{cond}}(\mathcal{C}), t)\|^2 + \lambda_2(t)\|\boldsymbol{\varepsilon} - \boldsymbol{\varepsilon}_\theta(\mathbf{X}_t, \mathbf{M}^{\mathrm{cond}}(\mathcal{C}, \mathcal{R}), t)\|^2], \quad (6)$$

where $\lambda_1(t)$ and $\lambda_2(t)$ are hyper-parameters. Note that both $\mathbf{M}^{\mathrm{cond}}(\mathcal{C})$ and $\mathbf{M}^{\mathrm{cond}}(\mathcal{C}, \mathcal{R})$ are derived from $\mathbf{X}_0$ via pre-designed chord recognition and rhythmic identification algorithms.

The guided noise prediction at timestep $t$ is then computed as

$$
\begin{aligned}
\boldsymbol{\varepsilon}_\theta(\mathbf{X}_t, t | \mathcal{C}, \mathcal{R}) =& \boldsymbol{\varepsilon}_\theta(\mathbf{X}_t, \boldsymbol{M}^{\text{cond}}(\mathcal{C}), t) \\
&+ w \cdot \left[ \boldsymbol{\varepsilon}_\theta(\mathbf{X}_t, \boldsymbol{M}^{\text{cond}}(\mathcal{C}, \mathcal{R}), t) - \boldsymbol{\varepsilon}_\theta(\mathbf{X}_t, \boldsymbol{M}^{\text{cond}}(\mathcal{C}), t) \right],
\end{aligned}
\tag{7}
$$

where $w$ is the weight parameter. Note that the general formulation $\boldsymbol{\varepsilon}_\theta(\mathbf{X}_t, t | \mathcal{C}, \mathcal{R})$ includes the case where rhythmic guidance is not provided ($\mathcal{R} = \emptyset$), and $w$ in equation 7 is set as 0.

## 4.2 FINE-GRAINED CONTROL IN SAMPLING PROCESS

As indicated by discussions in section 3.1 and empirical observations, providing chord conditions to model cannot prevent them from generating unwanted notes. Likewise, the rhythmic conditions also do not guarantee precise alignment with the provided rhythm. Therefore, in this section, we design a fine-grained sampling control to enhance the precision of generation.

We now introduce our method, which achieves precise harmonic control. Such control can be applied to serve two primary purposes: (1) eliminating out-of-key notes to enhance the reliability of the model's output and (2) shaping the output to reflect a specific tonal quality, such as that of the Dorian mode, by applying a tailored key sequence[6]. Notably, (2) does not require any training samples to be in the desired mode, as our harmonic control enables the model to adapt to tonal frameworks absent from the training data, without significantly disrupting the learned patterns.

Given key signature sequence $\mathcal{K}$ that aligns with chord condition $\mathcal{C}$, let $\omega_\mathcal{K}(\boldsymbol{l}) := \{l, \omega_\mathcal{K}(l)\}_{l=1}^L$ and denote all out-of-key positions implied by $\mathcal{K}$, the generated piano-roll $\widehat{\mathbf{M}}$ is expected to satisfy $\widehat{\mathbf{M}} \in \{0,1\}^{L \times H} \backslash \mathbb{W}_\mathcal{K}$, i.e., $\widehat{\mathbf{M}}_{lh} = 0$, for all $(l, h) \in \omega_\mathcal{K}(\boldsymbol{l})$. In other words, the desired constrained distribution for generated $\widehat{X}_0$ satisfies

$$
P\left( \widehat{\mathbf{X}}_0 \in \mathbb{W}'_\mathcal{K} := \left\{ \boldsymbol{X} \middle| \exists (l, h) \in \omega_\mathcal{K}(\boldsymbol{l}), \text{ s.t. } X_{lh} > 1/2 \right\} \middle| \mathcal{K} \right) = 0.
\tag{8}
$$

Note that in the backward sampling equation 2 that derives $\boldsymbol{X}_{t-1}$ from $\boldsymbol{X}_t$, we have for the first term (Song et al., 2020a; Chung et al., 2022)

$$
\left( \frac{\boldsymbol{X}_t - \sqrt{1 - \bar{\alpha}_t} \widehat{\varepsilon}_\theta(\boldsymbol{X}_t, t)}{\sqrt{\bar{\alpha}_t}} \right) = \text{"predicted } \mathbf{X}_0 \text{"} = \widehat{\mathbb{E}}[\mathbf{X}_0 | \boldsymbol{X}_t], \quad t = T, T-1, \ldots, 1.
\tag{9}
$$

The primary cause of generated out-of-key notes that fail to align with the context is the inaccurate estimation of the probability density $\widehat{p}_X$, which in turn affects the corresponding score function $\widehat{\boldsymbol{s}}_t(\boldsymbol{X}_t)$. The equivalence $\widehat{\boldsymbol{s}}_t(\boldsymbol{X}_t) = -\widehat{\varepsilon}_\theta(\boldsymbol{X}_t, t)/\sqrt{1 - \bar{\alpha}_t}$ therefore inspires us to project $\widehat{\mathbb{E}}[\mathbf{X}_0 | \boldsymbol{X}_t]$ to the $\mathcal{K}$-constrained domain $\mathbb{R}^{L \times H} \backslash \mathbb{W}'_\mathcal{K}$ by adjusting the value of $\widehat{\varepsilon}_\theta(\boldsymbol{X}_t, t)$ at every sampling step $t$. This adjustment is interpreted as an adjustment of the estimated score.

Specifically, using the notations in 4.1, at each sampling step $t$, we replace the guided noise prediction $\widehat{\varepsilon}_\theta(\boldsymbol{X}_t, t | \mathcal{C}, \mathcal{R})$ with $\tilde{\varepsilon}_\theta(\boldsymbol{X}_t, t | \mathcal{C}, \mathcal{R})$ such that

$$
\begin{aligned}
\tilde{\varepsilon}_\theta(\boldsymbol{X}_t, t | \mathcal{C}, \mathcal{R}) = \arg\min_{\boldsymbol{\varepsilon}} \quad & \| \boldsymbol{\varepsilon} - \widehat{\varepsilon}_\theta(\boldsymbol{X}_t, t | \mathcal{C}, \mathcal{R}) \| \\
\text{s.t.} \quad & \left( \frac{\boldsymbol{X}_t - \sqrt{1 - \bar{\alpha}_t} \boldsymbol{\varepsilon}}{\sqrt{\bar{\alpha}_t}} \right) \in \mathbb{R}^{L \times H} \backslash \mathbb{W}'_\mathcal{K}.
\end{aligned}
\tag{10}
$$

The element-wise formulation of $\tilde{\varepsilon}_\theta(\boldsymbol{X}_t, t | \mathcal{C}, \mathcal{R})$ is given as follows, with calculation details provided in Appendix B.2.

$$
\begin{aligned}
\tilde{\varepsilon}_{\theta,lh}(\boldsymbol{X}_t, t | \mathcal{C}, \mathcal{R}) = & \mathbf{1}\{(l, h) \notin \omega_\mathcal{K}(\boldsymbol{l})\} \cdot \widehat{\varepsilon}_{\theta,lh}(\boldsymbol{X}_t, t | \mathcal{C}, \mathcal{R}) \\
& + \mathbf{1}\{(l, h) \in \omega_\mathcal{K}(\boldsymbol{l})\} \cdot \max\left\{ \widehat{\varepsilon}_{\theta,lh}(\boldsymbol{X}_t, t | \mathcal{C}, \mathcal{R}), \frac{1}{\sqrt{1 - \bar{\alpha}_t}} \left( X_{t,lh} - \frac{\sqrt{\bar{\alpha}_t}}{2} \right) \right\}.
\end{aligned}
\tag{11}
$$

Plugging the adjusted noise prediction $\tilde{\varepsilon}_\theta(\boldsymbol{X}_t, t | \mathcal{C}, \mathcal{R})$ into equation 2, we derive the adjusted $\tilde{\mathbf{X}}_{t-1}$. The sampling process is therefore summarized as the following Algorithm 1.

---

[6]For example, the D Dorian scale consists of the pitch classes: D, E, F, G, A, B, and C

---

**Algorithm 1:** DDPM sampling with fine-grained harmonic control

---

**Input:** Input parameters: forward process variances $\beta_t$, $\bar{\alpha}_t = \prod_{s=1}^{t} \beta_t$, backward noise scale $\sigma_t$, chord condition $\mathcal{C}$, rhythmic condition $\mathcal{R}$ (can be null), key signature guidance $\mathcal{K}$

**Output:** generated piano roll $\tilde{\mathbf{M}} \in \{0,1\}^{L \times H}$

1 $\mathbf{X}_T \sim \mathcal{N}(0, \boldsymbol{I})$;

2 **for** $t = T, T-1, \ldots, 1$ **do**

3      Compute guided noise prediction $\widehat{\varepsilon}_\theta(\boldsymbol{X}_t, t | \mathcal{C}, \mathcal{R})$;

4      Perform noise correction: derive $\tilde{\varepsilon}_\theta(\boldsymbol{X}_t, t | \mathcal{C}, \mathcal{R})$ using equation 11;

5      Compute $\tilde{\mathbf{X}}_{t-1}$ by plugging the corrected noise $\tilde{\varepsilon}_\theta(\boldsymbol{X}_t, t | \mathcal{C}, \mathcal{R})$ into equation 2

6 **end**

7 Convert $\tilde{\mathbf{X}}_0$ into piano roll $\tilde{\mathbf{M}}$

8 **return** *output*;

---

Note that at the final step $t = 0$, the noise correction directly projects $\widehat{\mathbf{X}}_0$ to $\mathbb{R}^{L \times H} \backslash \mathbb{W}'_{\mathcal{K}}$, ensuring the probabilistic constraint 8. A natural concern is that enforcing precise fine-grained control over generated samples may disrupt the learned local patterns. The following proposition 2, proved in B.3, provides an upper bound that quantifies this potential effect and address the concern.

**Proposition 2.** *Under the SDE formulation in equation 3 and equation 4, given an early-stopping time $t_0$[7], if*

$$\mathbb{E}_{\mathbf{X}_t \sim p_t}[\|\varepsilon^*(\mathbf{X}_t, t) - \varepsilon_\theta(\mathbf{X}_t, t)\|^2] \leq \delta \qquad (12)$$

*for all $t$, where $\varepsilon^*(\mathbf{X}_t, t)$ is the optimal solution of the DDPM training objective (1), then we have*

$$KL(\tilde{p}_{t_0} | p_{t_0}) \leq \frac{\delta}{2} \int_{t_0}^{T} \frac{\beta(t)}{\sqrt{1 - e^{-\int_{t_0}^{t} \beta(s)ds}}} dt, \quad KL(\tilde{p}_{t_0} | \hat{p}_{t_0}) \leq \frac{\delta}{2} \int_{t_0}^{T} \frac{\beta(t)}{\sqrt{1 - e^{-\int_{t_0}^{t} \beta(s)ds}}} dt,$$

*where $p_{t_0}$ is the distribution of $\mathbf{X}_{t_0}$ in the forward process, $\hat{p}_{t_0}$ is the distribution of $\widehat{\mathbf{X}}_{t_0}$ generated by the diffusion sampling process without noise adjustment, and $\tilde{p}_{t_0}$ is the distribution of $\tilde{\mathbf{X}}_{t_0}$ generated by the fine-grained noise adjustment.*

Proposition 2 provides upper bounds for the distance between the controlled distribution and the uncontrolled distribution, as well as between the controlled distribution and the ground truth. We remark that, when applying an out-of-sample tonal framework control, such as using the Dorian scale as the key signature sequence $\mathcal{K}$ to shape the generated music towards the Dorian mode (a tonal framework not present in the training data), the generated distribution $\tilde{p}$ with fine-grained noise adjustment is fundamentally different from the ground truth distribution $p$. Nevertheless, Proposition 2 guarantees a substantial overlap between the two distributions $\tilde{p}$ and $p$, demonstrating a well-balanced interplay between external control and the model's internal learning from the training data, e.g., melodic lines. This theoretical insight aligns with our empirical observations, which is presented in the "Mode Change" section of the demo page.

## 5 EXPERIMENTS

In this section, we present experiments to demonstrate the effectiveness of our fine-grained guidance approach. We additionally create a demopage[8] for demonstration, which allows for fast and stable interactive music creation with user-specified input guidance, and even for generating music based on tonal frameworks absent from the training set.

### 5.1 NUMERICAL EXPERIMENTS

We present numerical experiments on accompaniment generation given both melody and chord generation, or symbolic music generation given only chord conditions. We focus on the former one

---

[7]We adopt the early-stopping time to avoid the blow-up of score function, which is standard in many literature (Song & Ermon, 2020; Nichol & Dhariwal, 2021)

[8]See https://huggingface.co/spaces/interactive-symbolic-music/InteractiveSymbolicMusicDemo. We note that slow performance may result from Huggingface resource limitations and network latency.

as it provides a more effective basis for comparison. Due to page limits, we put the results and more detailed explanation of the latter one in Appendix E.3. For the accompaniment generation task, we compare with two state-of-the-art baselines: 1) WholeSongGen (Wang et al. (2024)) and 2) GETMusic (Lv et al. (2023)).

### 5.1.1 Data Representation and Model Architecture

The generation target $X$ is represented by a piano-roll matrix of shape $2 \times L \times 128$ under the resolution of a 16th note, where $L$ represents the total length of the music piece, and the two channels represent note onset and sustain, respectively. In our experiments, we set $L = 64$, corresponding to a 4-measure piece under time signature $4/4$. Longer pieces can be generated autoregressively using the inpainting method. The backbone of our model is a 2D UNet with spatial attention.

The condition matrix $M^{\mathrm{cond}}$ is also represented by a piano roll matrix of shape $2 \times L \times 128$, with the same resolution and length as that of the generation target $X$. For the accompaniment generation experiments, we provide melody as an additional condition. Detailed construction of the condition matrices are provided in Appendix E.1.

### 5.1.2 Dataset

We use the POP909 dataset (Wang et al. (2020a)) for training and evaluation. This dataset consists of 909 MIDI pieces of pop songs, each containing lead melodies, chord progression, and piano accompaniment tracks. We exclude 29 pieces that are in triple meter. 90% of the data are used to train our model, and the remaining 10% are used for evaluation. In the training process, we split all the midi pieces into 4-measure non-overlapping segments (corresponding to $L = 64$ under the resolution of a 16th note), which in total generates 15761 segments in the entire training set. Training and sampling details are provided in Appendix E.2.

### 5.1.3 Task and Baseline Models

We consider accompaniment generation task based on melody and chord progression. We compare the performance of our model with two baseline models: 1) WholeSongGen (Wang et al. (2024)) and 2) GETMusic (Lv et al. (2023)). WholeSongGen is a hierarchical music generation framework that leverages cascaded diffusion models to generate full-length pop songs. It introduces a four-level computational music language, with the last level being accompaniment. The model for the last level can be directly used to generate accompaniment given music phrases, lead melody, and chord progression information. GETMusic is a versatile music generation framework that leverages a discrete diffusion model to generate tracks based on flexible source-target combinations. The model can also be directly applied to generate piano accompaniment conditioning on melody and chord. Since these baseline models do not support rhythm control, to ensure comparability, we will use the $M^{\mathrm{cond}}(\mathcal{C})$ without rhythm condition in our model.

### 5.1.4 Evaluation

We generate accompaniments for the 88 MIDI pieces in our evaluation dataset.[9] We introduce the following objective metrics to evaluate the generation quality of different methods:

*(1) Chord Progression Similarity* We use a rule-based chord recognition method from Dai et al. (2020) to recognize the chord progressions of the generated accompaniments and the ground truth accompaniments. Then we split all chord progressions into non-overlapping 2-measure segments, and encode each segment into a 256-d latent space use a pre-trained disentangled VAE (Wang et al. (2020b)). We then calculate the pairwise cosine similarities of the generated segments and the ground truth segments in the latent space. The average similarities with their 95% confidence intervals are shown in the first column of Table 1. The results indicate that our method significantly outperforms the other two baselines in chord accuracy.

*(2) Feature Distribution Overlapping Area* We assess the Overlapping Area (OA) of the distributions of some musical features in the generated and ground truth segments, including note pitch, duration,

---

[9]The WholeSongGen model from Wang et al. (2024) is also trained on the POP909 dataset. Our evaluation set is a subset of their test set so there is no in-sample evaluation issue on their model.

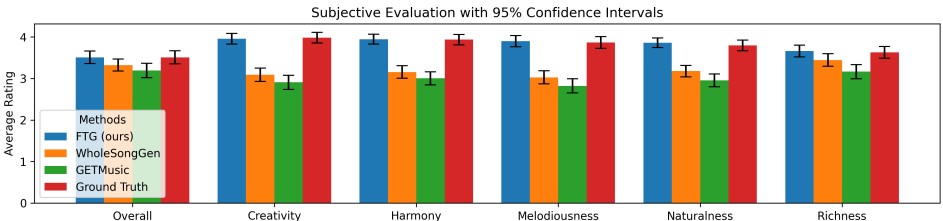

Figure 3: Subjective evaluation results on music quality.

and note density[10]. Similarly, we split both the generated accompaniments and the ground truth into non-overlapping 2-measure segments. Following von Rütte et al. (2023), for each feature $f$, we calculate the macro overlapping area (MOA) in segment-level feature distributions so that the metric also considers the temporal order of the features. MOA is defined as

$$MOA(f) = \frac{1}{N} \sum_{i=1}^{N} \text{overlap}(\pi_i^{\text{gen}}(f), \pi_i^{\text{gt}}(f)),$$

where $\pi_i^{\text{gen}}(f)$ is the distribution of feature $f$ in the $i$-th generated segment, and $\pi_i^{\text{gt}}(f))$ is the distribution of feature $f$ in the $i$-th ground truth segment. The MOA's for different methods are shown in the last 3 columns in Table 1. Again, our method significantly outperforms the baselines in terms of all the metrics.

| Methods | Chord Similarity | OA(pitch) | OA(duration) | OA(note density) |
|---------|------------------|-----------|--------------|------------------|
| **FTG (Ours)** | **$0.720 \pm 0.007$** | **$0.643 \pm 0.005$** | **$0.644 \pm 0.006$** | **$0.845 \pm 0.005$** |
| WholeSongGen | $0.611 \pm 0.010$ | $0.471 \pm 0.006$ | $0.586 \pm 0.005$ | $0.726 \pm 0.005$ |
| GETMusic | $0.394 \pm 0.012$ | $0.323 \pm 0.010$ | $0.377 \pm 0.011$ | $0.661 \pm 0.011$ |

Table 1: Evaluation of the similarity with ground truth for all methods.

*(3) Subjective Evaluation*

To compare performance of our FTG method against the baselines (ground truth, WholeSongGen, and GETMusic), we prepared 6 sets of generated samples, with each set containing the melody paired with accompaniments generated by FTG, WholeSongGen, and GETMusic, along with the ground truth accompaniment. This yields a total of $6 \times 4 = 24$ samples. The samples are presented in a randomized order, and their sources are not disclosed to participants. Experienced listeners assess the quality of samples in 5 dimensions: creativity, harmony (whether the accompaniment is in harmony with the melody), melodiousness, naturalness and richness, together with an overall assessment. The results are shown in Figure 3. The bar height shows the mean rating, and the error bar shows the 95% confidence interval. FTG consistently outperforms the baselines in all dimensions. For details of our survey, please see Appendix F.

### 5.1.5 ABLATION STUDY

In this section, we conduct ablation studies to better illustrate the effectiveness of our FTG method. We aim to demonstrate the effectiveness of both the fine-grained training condition and the sampling control. We also compare with the simple rule-based post-sample editing. The former leverages the structured gradual denoising process of diffusion models, ensuring a theoretical guarantee of preserving the distributional properties of the original learned distribution. In contrast, the latter employs a brute-force editing approach that disrupts the generated samples, affecting local melodic lines and rhythmic patterns. The numerical results further validate this analysis.

The specific experimental settings are given as follows: our first experiment involves the same model trained with fine-grained conditioning but only removes the out-of-key notes after the last sampling

---

[10]Note density is the number of onset notes at each time

step; the second also incorporates fine-grained conditioning for training but without any control during sampling; the third is an unconditional model without any conditioning or control in both the training and sampling process. All experiments use the same model architecture and random seeds as the one with full control for comparability.

We evaluate the frequency of out-of-key notes by computing the percentage of steps in the generated sequences containing at least one out-of-key note, where each step corresponds to a 16th note. Additionally, we assessed overall model performance using the same quantitative metrics as in the previous section. The results are shown in Table 2. To interpret, the fine-grained conditioning (i.e., training control) provides a great improvement in model performance, and adding sampling control can ensures further improvements. Moreover, while rule-based post-sampling editing achieves some improvement in pitch and chord similarity, it is still outperformed by our fine-grained sampling control method, Our method fully leverages the structured, gradual denoising process of diffusion models to guide the model in correcting or replacing incorrect notes, while preserving structures of the original learned distribution.

| Methods | % Out-of-Key Notes | Chord Similarity | OA (pitch) | OA (duration) | OA (note density) |
|---|---|---|---|---|---|
| **Training and Sampling Control** | 0.0% | **0.720** ±**0.007** | **0.643** ±**0.005** | **0.644** ±**0.006** | **0.845** ±**0.005** |
| Training Control Edit After Sampling | 0.0% | 0.712 ±0.007 | 0.631 ±0.005 | 0.643 ±0.005 | 0.835 ±0.003 |
| Only Training Control | 6.0% | 0.690 ±0.008 | 0.614 ±0.005 | 0.643 ±0.005 | 0.829 ±0.004 |
| No Control | 10.1% | 0.378 ±0.007 | 0.427 ±0.006 | 0.265 ±0.007 | 0.682 ±0.005 |

Table 2: Comparison of the results with and without control in the sampling process.

## 5.2 EMPIRICAL OBSERVATIONS

Notably, harmonic control not only helps the model eliminate incorrect notes, but also guides it to replace them with correct ones. Such representative examples are presented in Appendix G. Our demo page contains the following parts:

- Samples of diffusion models without sampling control that include dissonant out-of-key notes, demonstrating the challenge in precision and underscoring the value of effective sampling control.

- Samples of accompaniment generation results of our model

- Samples of symbolic music generated in the Dorian scale and the Chinese pentatonic scale, illustrating their respective tonal characteristics and musical frameworks.

- A user-interface that allows real-time conditional accompaniment generation with melody and chord conditions

## 6 CONCLUSION

In this work, we apply fine-grained textural guidance (FTG) on symbolic music generation models. We provide theoretical analysis and empirical evidence to highlight the need for fine-grained and precise control over the model output. We also provide theoretical analysis to quantify and upper bound the potential effect of fine-grained control on learned local patterns, and provide samples and numerical results for demonstrating the effectiveness of our approach. For the impact of our method, we note that the FTG method can be integrated with other diffusion-based symbolic music generation methods. While sacrificing some creative flexibility, the FTG method prioritizes real-time generation stability and enables efficient generation with precise control.

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

# A   SUMMARY OF NON-STANDARD NOTATIONS

Table 3: Summary of Notations

| Notation | Type | Description |
|---|---|---|
| **Piano Roll-like Matrices** | | |
| $\boldsymbol{M}, \mathbf{M}, \mathbf{M}^i$ | discrete | Fixed/random/samples of piano roll in $\{0,1\}^{L \times H}$. |
| $\widehat{\mathbf{M}}$ | discrete | Estimated or generated piano roll |
| $\tilde{\mathbf{M}}$ | discrete | Generated piano roll with fine-grained sampling guidance |
| $\boldsymbol{M}^{\text{cond}}$ | discrete | The piano roll representing fine-grained conditions. |
| $\boldsymbol{X}, \mathbf{X}, \mathbf{X}^i$ | continuous | Fixed/random/samples of continuous approximation of piano roll. |
| $\widehat{\mathbf{X}}, \widehat{\mathbf{X}}_0$ | continuous | Estimated or generated value of $\mathbf{X}$, diffusion output. |
| $\tilde{\mathbf{X}}_0, \tilde{\mathbf{X}}_t$ | continuous | Diffusion samples with fine-grained sampling guidance |
| **Textural Conditions or Guidance** (abstract) | | |
| $\mathcal{K}, \mathcal{K}(l)$ | condition/control | Key-signature condition or control for entire piano roll/at time $l$. |
| $\mathcal{C}, \mathcal{C}(l)$ | condition | Chord condition. |
| $\mathcal{R}, \mathcal{R}(l)$ | condition | Rhythmic condition. |
| $\mathcal{B}, \mathcal{B}(l)$ | control | Rhythmic control. |
| **Set of Indexes** | | |
| $\boldsymbol{l}$ | $\subset [\![1, L]\!]$ | set of time values. |
| $\boldsymbol{h}, \boldsymbol{h}(l)$ | $\subset [\![1, H]\!]$ | set of pitch values (as function of $l$). |
| $\omega_{\mathcal{K}}(l)$ | $\subset [\![1, H]\!]$ | pitch values that are out of key $\mathcal{K}$ at time $l$. |
| $\gamma_{\mathcal{C}}(l)$ | $\subset [\![1, H]\!]$ | pitch values corresponding to chord $\mathcal{C}(l)$ at time $l$. |
| $\gamma_{\mathcal{R}}$ | $\subset [\![1, L]\!]$ | onset time values corresponding to rhythm $\mathcal{R}$. |
| **Set of Matrices** | | |
| $\mathbb{W}_{\mathcal{K}}$ | set of $\boldsymbol{M}$ | with out-of-key notes for key signature $\mathcal{K}$. |
| $\mathbb{W}'_{\mathcal{K}}$ | set of $\boldsymbol{X}$ | corresponding to set $\mathbb{W}_{\mathcal{K}}$ of $\boldsymbol{M}$. |
| $\mathbb{C}_{\mathcal{K}}$ | set of $\boldsymbol{M}$ | with contexts accommodating out-of-key $\mathcal{K}$ notes. |
| **Probability and Events** | | |
| $\boldsymbol{w}, \boldsymbol{w}_1$ | event | $\mathbf{M}$ has out-of-key notes. |
| $\boldsymbol{c}, \bar{\boldsymbol{c}}$ | event | $\mathbf{M}$ has or does not have "good contexts" |
| $P_{\mathbf{M}}, \widehat{P}_{\mathbf{M}}$ | discrete probability | Probability/estimated probability regarding distribution of $\mathbf{M}$ |
| $p_{\mathbf{X}}, \widehat{p}_{\mathbf{X}}$ | density | Density/estimated density regarding distribution of $\mathbf{X}$ |
| $\mathcal{P}, \mathcal{P}_{\delta}$ | class | Distribution class of $p_{\mathbf{X}}$ |

## B  Proof of propositions and calculation details

### B.1  Proof of proposition 1

We first provide the following definition 1, which is adopted from Fu et al. (2024).

**Definition 1.** *Denote the space of density functions*

$$\mathcal{P}_0 = \left\{ p(\boldsymbol{X}) = f(\boldsymbol{X}) \exp(-C\|\boldsymbol{X}\|_2^2) : f \in \mathcal{L}(\mathbb{R}^{L \times H}, B), f(\boldsymbol{X}) \geq \alpha > 0 \right\},$$

*where $C$ and $\alpha$ can be any given constants, and $\mathcal{L}(\mathbb{R}^{L \times H}, B)$ denotes the class of Lipschitz continuous functions on $\mathbb{R}^{L \times H}$ with Lipschitz constant bounded by $B$.*

*Suppose that the density function of $\mathbf{X}$ belongs to the following space*

$$\mathcal{P}_\delta = \{ p(\boldsymbol{X}) \in \mathcal{P}_0 | P_{\boldsymbol{M}}(\bar{\boldsymbol{c}}, \boldsymbol{w}) = \delta \}, \tag{13}$$

*where the distribution of $\mathbf{M}$ is defined from $\mathbf{X}$ by*

$$\mathrm{M}_{lh} = \mathbf{1}\{\mathrm{X}_{lh} \geq 1/2\}.$$

**Proposition 3.** *Consider approximating $P_{\mathbf{M}}$ with the distribution of a continuous random variable $\mathbf{X}$. Suppose $n$ i.i.d. data $\{\mathbf{X}^i\}_{i=1}^n$ come from distribution $p_{\mathbf{X}}$. Let $\{\mathbf{M}^i\}_{i=1}^n$ where $\mathrm{M}_{lh}^i = \mathbf{1}\{\mathrm{X}_{lh}^i \geq 1/2\}$ be the training data provided to the continuous estimator $\widehat{p}_{\mathbf{X}}$. Let $\widehat{P}_{\mathbf{M}}$ be derived from $\widehat{p}_{\mathbf{X}}$ via the connection $\widehat{\mathrm{M}}_{lh}^i = \mathbf{1}\{\widehat{\mathrm{X}}_{lh}^i \geq 1/2\}$. We have $\exists C > 0$,*

$$\inf_{\widehat{p}_{\mathbf{X}}} \sup_{p_{\mathbf{X}} \in \mathcal{P}_\delta} \mathbb{E}_{\{\mathbf{M}^i\}_{i=1}^n} \widehat{P}_{\mathbf{M}}(\bar{\boldsymbol{c}}, \boldsymbol{w}) \geq C \cdot n^{-\frac{1}{LH+2}} - P_{\mathbf{M}}(\bar{\boldsymbol{c}}, \boldsymbol{w}), \tag{14}$$

*where $\widehat{P}_{\mathbf{M}}$ is derived from $\widehat{p}_{\mathbf{X}}$ via the connection $\widehat{\mathrm{M}}_{lh}^i = \mathbf{1}\{\widehat{\mathrm{X}}_{lh}^i \geq 1/2\}$.*

*Proof.* We first restate a special case of proposition 4.3 of Fu et al. (2024) as the following lemma.

**Lemma 1.** *(Fu et al. (2024), proposition 4.3) Fix a constant $C_2 > 0$. Consider estimating a distribution $P(\mathbf{x})$ with a density function belonging to the space*

$$\mathcal{P} = \left\{ p(\mathbf{x}) = f(\mathbf{x}) \exp(-C_2\|\mathbf{x}\|_2^2) : f(\mathbf{x}) \in \mathcal{L}(\mathbb{R}^d, B), f(\mathbf{x}) \geq C > 0 \right\}.$$

*Given $n$ i.i.d. data $\{x_i\}_{i=1}^n$, we have*

$$\inf_{\hat{\mu}} \sup_{p \in \mathcal{P}} \mathbb{E}_{\{x_i\}_{i=1}^n} \left[ TV(\hat{\mu}, P) \right] \gtrsim n^{-\frac{1}{d+2}},$$

*where the infimum is taken over all possible estimators $\hat{\mu}$ based on the data.*

From lemma 1, since all the conditions are satisfied, we know that

$$\inf_{\widehat{p}_{\mathbf{X}}} \sup_{p_{\mathbf{X}} \in \mathcal{P}_0} \mathbb{E}_{\{x_i\}_{i=1}^n} \left[ \mathrm{TV}(\widehat{p}_{\mathbf{X}}, p_{\mathbf{X}}) \right] \gtrsim n^{-\frac{1}{LH+2}}, \tag{15}$$

where

$$\mathrm{TV}(\widehat{p}_{\mathbf{X}}, p_{\mathbf{X}}) = \int_{\mathbb{R}^{L \times H}} |\widehat{p}_{\mathbf{X}}(\boldsymbol{X}) - p_{\mathbf{X}}(\boldsymbol{X})| d\boldsymbol{X}. \tag{16}$$

From the following, all distribution and density functions are conditional distributions and densities with key signature condition $\mathcal{K}$, therefore, we omit the term $\mathcal{K}$ for simplicity of notations.

Without loss of generality, suppose event $\boldsymbol{w}_1$ denoting a note-out-of-key occurring at $(l, h) = (1, 1)$ is contained in $\boldsymbol{w}$. By $P_{\mathbf{M}}(\bar{\boldsymbol{c}}, \boldsymbol{w}) = 0$, we have

$$\begin{aligned} \widehat{P}_{\mathbf{M}}(\boldsymbol{w}_1) &= \int_{(\frac{1}{2}, +\infty)} dX_{11} \int_{\mathbb{R}^{L \times H - 1}} d\boldsymbol{Y} \, \widehat{p}_{\mathbf{X}}(X_{11}, \boldsymbol{Y}) \\ &\triangleq \int_{\Omega_{\boldsymbol{w}_1}} \widehat{p}_{\mathbf{X}}(\boldsymbol{X}) d\boldsymbol{X}, \end{aligned} \tag{17}$$

where $\boldsymbol{Y}$ is a $(LH - 1)$-dimensional variable denoting the elements in matrix $\boldsymbol{X}$ excluding $X_{11}$. Let $\mathbb{C}(\boldsymbol{w}_1)$ denotes the set of all possible realizations of piano roll $\boldsymbol{M}$ with a "good context" to accommodate the out-of-key note $\boldsymbol{w}_1$, and contains the note $\boldsymbol{w}_1$. For each $\boldsymbol{M} \in \mathbb{C}(\boldsymbol{w}_1)$, let

$$\delta(\boldsymbol{M}) = \{(l, h) \in [\![1, L]\!] \times [\![1, H]\!] | M_{lh} = 1\}.$$

We have

$$\begin{aligned}
\widehat{P}_{\mathbf{M}}(\boldsymbol{c}, \boldsymbol{w}_1) &= \sum_{\boldsymbol{M} \in \mathbb{C}^{\boldsymbol{w}_1}} \int_{(\frac{1}{2}, +\infty)^{|\delta(\boldsymbol{M})|}} dX_{\delta(\boldsymbol{M})} \int_{(-\infty, \frac{1}{2})^{L \times H - |\delta(\boldsymbol{M})|}} d\boldsymbol{Y} \, \widehat{p}_{\mathbf{X}}(X_{\delta(\boldsymbol{M})}, X_{L \times H \setminus \delta(\boldsymbol{M})}) \\
&\triangleq \int_{\Omega_{\mathbb{C}(\boldsymbol{w}_1)}} \widehat{p}_{\mathbf{X}}(\boldsymbol{X}) d\boldsymbol{X},
\end{aligned}$$

(18)

and note that $\Omega_{\mathbb{C}(\boldsymbol{w}_1)} \subset \Omega_{\boldsymbol{w}_1}$, we have

$$\widehat{P}_{\mathbf{M}}(\bar{\boldsymbol{c}}, \boldsymbol{w}_1) = \widehat{P}_{\mathbf{M}}(\boldsymbol{w}_1) - \widehat{P}_{\mathbf{M}}(\boldsymbol{c}, \boldsymbol{w}_1) = \int_{\Omega_{\boldsymbol{w}_1} \setminus \Omega_{\mathbb{C}(\boldsymbol{w}_1)}} \widehat{p}_{\mathbf{X}}(\boldsymbol{X}) d\boldsymbol{X}$$

(19)

To better explain and summarize equation 17, equation 18 and equation 19, $\widehat{P}_{\mathbf{M}}(\cdot)$ is always calculated by integrating $\widehat{p}_{\mathbf{X}}(\boldsymbol{X})$ on a corresponding domain. Similarly, for the ground truth distributions and under definition 1 which provides $P_{\boldsymbol{M}}(\bar{\boldsymbol{c}}, \boldsymbol{w}) = \delta$, we have

$$P_{\boldsymbol{M}}(\bar{\boldsymbol{c}}, \boldsymbol{w}_1) = \int_{\Omega_{\boldsymbol{w}_1} \setminus \Omega_{\mathbb{C}(\boldsymbol{w}_1)}} p_{\mathbf{X}}(\boldsymbol{X}) d\boldsymbol{X} \leq \delta.$$

Therefore,

$$\begin{aligned}
\widehat{P}_{\mathbf{M}}(\bar{\boldsymbol{c}}, \boldsymbol{w}_1) &= \int_{\Omega_{\boldsymbol{w}_1} \setminus \Omega_{\mathbb{C}(\boldsymbol{w}_1)}} \widehat{p}_{\mathbf{X}}(\boldsymbol{X}) d\boldsymbol{X} \\
&\geq \int_{\Omega_{\boldsymbol{w}_1} \setminus \Omega_{\mathbb{C}(\boldsymbol{w}_1)}} |\widehat{p}_{\mathbf{X}}(\boldsymbol{X}) - p_{\mathbf{X}}(\boldsymbol{X})| - p_{\mathbf{X}}(\boldsymbol{X}) d\boldsymbol{X} \\
&\geq \int_{\Omega_{\boldsymbol{w}_1} \setminus \Omega_{\mathbb{C}(\boldsymbol{w}_1)}} |\widehat{p}_{\mathbf{X}}(\boldsymbol{X}) - p_{\mathbf{X}}(\boldsymbol{X})| \, d\boldsymbol{X} - \delta
\end{aligned}$$

(20)

Therefore,

$$\widehat{P}_{\mathbf{M}}(\bar{\boldsymbol{c}}, \boldsymbol{w}_1) = \text{TV}|_{\Omega_{\boldsymbol{w}_1} \setminus \Omega_{\mathbb{C}(\boldsymbol{w}_1)}}(\widehat{p}_{\mathbf{X}}, p_{\mathbf{X}}) - \delta,$$

(21)

where $\text{TV}|_{\Omega_{\boldsymbol{w}_1} \setminus \Omega_{\mathbb{C}(\boldsymbol{w}_1)}}$ is the total variation integral restricted on the domain $\Omega_{\boldsymbol{w}_1} \setminus \Omega_{\mathbb{C}(\boldsymbol{w}_1)}$.

By construction of packing numbers provided in the proof of proposition 4.3 of Fu et al. (2024), we note that constraint $P_{\boldsymbol{M}}(\bar{\boldsymbol{c}}, \boldsymbol{w}) = \delta$ or restricting the integral of total variation on $\Omega_{\boldsymbol{w}_1} \setminus \Omega_{\mathbb{C}(\boldsymbol{w}_1)}$ does not change the order of the packing numbers, i.e., $\mathcal{P}_0$ and $\mathcal{P}_\delta$ have the same packing numbers. Let

$$\mathcal{P}_\delta^{\Omega_{\boldsymbol{w}_1} \setminus \Omega_{\mathbb{C}(\boldsymbol{w}_1)}} = \left\{ C(\Omega_{\boldsymbol{w}_1} \setminus \Omega_{\mathbb{C}(\boldsymbol{w}_1)}) \cdot p(\mathbf{X}) \mathbf{1}_{\mathbf{X} \in \Omega_{\boldsymbol{w}_1} \setminus \Omega_{\mathbb{C}(\boldsymbol{w}_1)}} \mid p(\mathbf{X}) \in \mathcal{P}_\delta \right\},$$

where the constant $C(\Omega_{\boldsymbol{w}_1} \setminus \Omega_{\mathbb{C}(\boldsymbol{w}_1)})$ is a scale factor to ensure that $C(\Omega_{\boldsymbol{w}_1} \setminus \Omega_{\mathbb{C}(\boldsymbol{w}_1)}) \cdot p(\mathbf{X}) \mathbf{1}_{\mathbf{X} \in \Omega_{\boldsymbol{w}_1} \setminus \Omega_{\mathbb{C}(\boldsymbol{w}_1)}}$ is a probability density function. For simplicity we use $\mathcal{P}(\delta, \boldsymbol{w}_1)$ for short of $\mathcal{P}_\delta^{\Omega_{\boldsymbol{w}_1} \setminus \Omega_{\mathbb{C}(\boldsymbol{w}_1)}}$.

We have

$$\inf_{\widehat{p}_{\mathbf{X}}} \sup_{p \in \mathcal{P}(\delta, \mathbf{w}1)} \mathbb{E}_{\{\mathbf{X}_i\}_{i=1}^n} \text{TV}(\widehat{p}_{\mathbf{X}}, p_{\mathbf{X}}) \gtrsim n^{-\frac{1}{LH+2}}.$$

(22)

Combining with equation 21, and noting that $\widehat{P}_{\mathbf{M}}(\bar{\boldsymbol{c}}, \boldsymbol{w}) \geq \widehat{P}_{\mathbf{M}}(\bar{\boldsymbol{c}}, \boldsymbol{w}_1)$, we have

$$\begin{aligned}
\inf_{\widehat{p}_{\mathbf{X}}} \sup_{p \in \mathcal{P}_\delta} \mathbb{E}_{\{\mathbf{X}_i\}_{i=1}^n} \widehat{P}_{\mathbf{M}}(\bar{\boldsymbol{c}}, \boldsymbol{w}) + \delta &= \inf_{\widehat{p}_{\mathbf{X}}} \sup_{p \in \mathcal{P}_\delta} \text{TV}|_{\Omega_{\boldsymbol{w}_1} \setminus \Omega_{\mathbb{C}(\boldsymbol{w}_1)}}(\widehat{p}_{\mathbf{X}}, p_{\mathbf{X}}) - \delta \\
\inf_{\widehat{p}_{\mathbf{X}}} \sup_{p \in \mathcal{P}(\delta, \mathbf{w}1)} &\geq \text{TV}(\widehat{p}_{\mathbf{X}}, p_{\mathbf{X}}) \gtrsim n^{-\frac{1}{LH+2}}.
\end{aligned}$$

Therefore, $\exists C > 0, \forall n$,

$$\inf_{\widehat{p}_{\mathbf{X}}} \sup_{p \in \mathcal{P}_\delta} \mathbb{E}_{\{\mathbf{X}_i\}_{i=1}^n} \widehat{P}_{\mathbf{M}}(\bar{\boldsymbol{c}}, \boldsymbol{w}) \geq C \cdot n^{-\frac{1}{LH+2}} - P_{\boldsymbol{M}}(\bar{\boldsymbol{c}}, \boldsymbol{w}).$$

which finishes the proof. $\qquad \square$

## B.2 CALCULATION DETAILS IN 4.2

Our goal is to find the optimal solution of problem (10). Since the constraint is an element-wise constraint on a linear function of $\varepsilon$ and the objective is separable, we can find the optimal solution by element-wise optimization. Consider the $(l, h)$-element of $\varepsilon$.

First, if $(l, h) \notin \omega_{\mathcal{K}}(l)$, there is no constraint on $\varepsilon_{lh}$. Therefore, the optimal solution of $\varepsilon_{lh}$ is $\widehat{\varepsilon}_{\theta,lh}(\boldsymbol{X}_t, t | \mathcal{C}, \mathcal{R})$.

If $(l, h) \in \omega_{\mathcal{K}}(l)$, the constraint on $\varepsilon_{lh}$ is

$$X_{t,lh} - \frac{\sqrt{1 - \bar{\alpha}_t}\varepsilon_{lh}}{\sqrt{\bar{\alpha}_t}} \leq \frac{1}{2},$$

which is equivalent to

$$\varepsilon_{lh} \geq \frac{1}{\sqrt{1 - \bar{\alpha}_t}} \left( X_{t,lh} - \frac{\sqrt{\bar{\alpha}_t}}{2} \right).$$

The objective is to minimize $\|\varepsilon_{lh} - \widehat{\varepsilon}_{\theta,lh}(\boldsymbol{X}_t, t | \mathcal{C}, \mathcal{R})\|$. Therefore, the optimal solution of $\varepsilon_{lh}$ is

$$\varepsilon_{lh} = \max \left\{ \widehat{\varepsilon}_{\theta,lh}(\boldsymbol{X}_t, t | \mathcal{C}, \mathcal{R}), \frac{1}{\sqrt{1 - \bar{\alpha}_t}} \left( X_{t,lh} - \frac{\sqrt{\bar{\alpha}_t}}{2} \right) \right\}.$$

## B.3 PROOF OF PROPOSITION 2

*Proof.* Recall that According to Song et al. (2020b), the DDPM forward process $\mathbf{X}_t = \sqrt{\bar{\alpha}_t}\mathbf{X}_0 + \sqrt{1 - \bar{\alpha}_t}\varepsilon$ can be regarded as a discretization of the following SDE:

$$d\mathbf{X}_t = -\frac{1}{2}\beta(t)\mathbf{X}_t dt + \sqrt{\beta(t)}d\mathbf{W}_t,$$

and the corresponding denoising process takes the form of a solution to the following stochastic differential equation (SDE):

$$d\mathbf{X}_t = - \left[ \frac{1}{2}\beta(t)\mathbf{X}_t + \beta(t)\nabla_{\mathbf{X}_t} \log p_t(\mathbf{X}_t) \right] dt + \sqrt{\beta(t)}d\bar{\mathbf{W}}_t,$$

where $\beta(t/T) = T\beta_t$ as $T$ goes to infinity, $\bar{\mathbf{W}}_t$ is the reverse time standard Wiener process, and $\bar{\alpha}_t$ term should be replaced by its continuous version $e^{-\int_0^t \beta(s)ds}$ (or $e^{-\int_{t_0}^t \beta(s)ds}$ when early-stopping time $t_0$ is adopted). The score function $\nabla_{\mathbf{X}_t} \log p_t(\mathbf{X}_t)$ can be approximated by $-\varepsilon_\theta(\mathbf{X}_t, t)/\sqrt{1 - e^{-\int_0^t \beta(s)ds}}$.

Under the SDE formulation, the denoising process can take the form of a solution to stochastic differential equation (SDE):

$$d\mathbf{X}_t = - \left[ \frac{1}{2}\beta(t)\mathbf{X}_t + \beta(t)\nabla_{\mathbf{X}_t} \log p_t(\mathbf{X}_t) \right] dt + \sqrt{\beta(t)}d\bar{\mathbf{W}}_t, \tag{23}$$

where $\beta(t/T) = T\beta_t$, $\bar{\mathbf{W}}_t$ is the reverse time standard Wiener process. According to Song et al. (2020b), as $T \to \infty$, the solution to the SDE converges to the real data distribution $p_0$.

In the diffusion model, $\nabla_{\mathbf{X}_t} \log p_t(\mathbf{X}_t)$ is approximated by $-\varepsilon_\theta(\mathbf{X}_t, t)/\sqrt{1 - e^{-\int_{t_0}^t \beta(s)ds}}$. Therefore, the approximated reverse-SDE sampling process without harmonic guidance is

$$d\hat{\mathbf{X}}_t = - \left[ \frac{1}{2}\beta(t)\hat{\mathbf{X}}_t - \beta(t)\frac{\varepsilon_\theta(\hat{\mathbf{X}}_t, t)}{\sqrt{1 - e^{-\int_{t_0}^t \beta(s)ds}}} \right] dt + \sqrt{\beta(t)}d\bar{\mathbf{W}}_t. \tag{24}$$

Similarly, the sampling process with fine-grained harmonic guidance is

$$d\tilde{\mathbf{X}}_t = - \left[ \frac{1}{2}\beta(t)\tilde{\mathbf{X}}_t - \beta(t)\frac{\tilde{\varepsilon}_\theta(\tilde{\mathbf{X}}_t, t)}{\sqrt{1 - e^{-\int_{t_0}^t \beta(s)ds}}} \right] dt + \sqrt{\beta(t)}d\bar{\mathbf{W}}_t, \tag{25}$$

where $\tilde{\varepsilon}_\theta$ is defined as equation 10 and equation 11.

For simplicity, we denote the drift terms as follows:

$$f(\mathbf{X}_t, t) = -\left[\frac{1}{2}\beta(t)\mathbf{X}_t + \beta(t)\nabla_{\mathbf{X}_t}\log p_t(\mathbf{X}_t)\right]$$

$$\hat{f}(\hat{\mathbf{X}}_t, t) = -\left[\frac{1}{2}\beta(t)\hat{\mathbf{X}}_t - \beta(t)\frac{\varepsilon_\theta(\hat{\mathbf{X}}_t, t)}{\sqrt{1 - e^{-\int_{t_0}^t \beta(s)ds}}}\right],$$

$$\tilde{f}(\tilde{\mathbf{X}}_t, t) = -\left[\frac{1}{2}\beta(t)\tilde{\mathbf{X}}_t - \beta(t)\frac{\tilde{\varepsilon}_\theta(\tilde{\mathbf{X}}_t, t)}{\sqrt{1 - e^{-\int_{t_0}^t \beta(s)ds}}}\right].$$

Since

$$\mathbb{E}_{\mathbf{X}_t \sim p_t}[\|\varepsilon^*(\mathbf{X}_t, t) - \varepsilon_\theta(\mathbf{X}_t, t)\|^2] \leq \delta,$$

and

$$\varepsilon^*(\mathbf{X}_t, t) = -\sqrt{1 - e^{-\int_{t_0}^t \beta(s)ds}}\nabla_{\mathbf{X}_t}\log p_t(\mathbf{X}_t),$$

we have

$$\mathbb{E}_{\mathbf{X} \sim p_t}[\|f(\mathbf{X}, t) - \hat{f}(\mathbf{X}, t)\|] \leq \frac{\beta(t)}{\sqrt{1 - e^{-\int_{t_0}^t \beta(s)ds}}}\delta.$$

Now we consider $\tilde{\varepsilon}_\theta(\tilde{\mathbf{X}}_t, t)$, which is the solution of the optimization problem (10). In the continuous SDE case, the corresponding optimization problem becomes

$$\min_{\varepsilon} \quad \|\varepsilon - \hat{\varepsilon}_\theta(\mathbf{X}_t, t | \mathcal{C}, \mathcal{R})\|$$

$$\text{s.t.} \quad \left(\frac{\mathbf{X}_t - \sqrt{1 - e^{-\int_{t_0}^t \beta(s)ds}}\varepsilon}{e^{-\frac{1}{2}\int_{t_0}^t \beta(s)ds}}\right) \in \mathbb{R}^{L \times H}\backslash\mathbb{W}'_\mathcal{K}. \quad (26)$$

According to Proposition 1 of Chung et al. (2022), the posterior mean of $\mathbf{X}_0$ conditioning on $\mathbf{X}_t$ is

$$\mathbb{E}[\mathbf{X}_0 | \mathbf{X}_t] = \frac{1}{e^{-\frac{1}{2}\int_{t_0}^t \beta(s)ds}}\left(\mathbf{X}_t + (1 - e^{-\frac{1}{2}\int_{t_0}^t \beta(s)ds})\nabla_{\mathbf{X}_t}\log p_t(\mathbf{X}_t)\right)$$

$$= \frac{1}{e^{-\frac{1}{2}\int_{t_0}^t \beta(s)ds}}\left(\mathbf{X}_t - \sqrt{1 - e^{-\int_{t_0}^t \beta(s)ds}}\varepsilon^*(\mathbf{X}_t, t)\right).$$

Since the domain of $\mathbf{X}_0$ is $R^{L \times H}\backslash\mathbb{W}'_\mathcal{K}$, which is a convex set, we know that the posterior mean $\mathbb{E}[\mathbf{X}_0 | \mathbf{X}_t]$ naturally belongs to its domain. Therefore, $\varepsilon^*(\mathbf{X}_t, t)$ is feasible to the problem (26). Since the optimal solution of the problem is $\tilde{\varepsilon}_\theta(\mathbf{X}_t, t)$, we have

$$\|\tilde{\varepsilon}_\theta(\mathbf{X}_t, t) - \varepsilon_\theta(\mathbf{X}_t, t)\| \leq \|\varepsilon^*(\mathbf{X}_t, t) - \varepsilon_\theta(\mathbf{X}_t, t)\|$$

for all $\mathbf{X}_t$ and $t$. This further leads to the result that

$$\mathbb{E}_{\mathbf{X} \sim p_t}[\|\tilde{f}(\mathbf{X}, t) - \hat{f}(\mathbf{X}, t)\|] \leq \frac{\beta(t)}{\sqrt{1 - e^{-\int_{t_0}^t \beta(s)ds}}}\delta. \quad (27)$$

Moreover, since $\tilde{\varepsilon}_\theta(\mathbf{X}_t, t)$ is essentially the projection of $\varepsilon_\theta(\mathbf{X}_t, t)$ onto the convex set defined by the constraints in (26), and $\varepsilon^*(\mathbf{X}_t, t)$ also belongs to the set, we know that the inner product of $\varepsilon^*(\mathbf{X}_t, t) - \tilde{\varepsilon}_\theta(\mathbf{X}_t, t)$ and $\varepsilon_\theta(\mathbf{X}_t, t) - \tilde{\varepsilon}_\theta(\mathbf{X}_t, t)$ is negative, which further leads to the result that

$$\|\tilde{\varepsilon}_\theta(\mathbf{X}_t, t) - \varepsilon^*(\mathbf{X}_t, t)\| \leq \|\varepsilon^*(\mathbf{X}_t, t) - \varepsilon_\theta(\mathbf{X}_t, t)\|, \quad (28)$$

which further implies

$$\mathbb{E}_{\mathbf{X} \sim p_t}[\|\tilde{f}(\mathbf{X}, t) - f(\mathbf{X}, t)\|] \leq \frac{\beta(t)}{\sqrt{1 - e^{-\int_{t_0}^t \beta(s)ds}}}\delta. \quad (29)$$

The following Girsanov's Theorem (Karatzas & Shreve (1991)) will be used (together with equation 27 and equation 29) to prove the upper bounds for the KL-divergences in our Proposition 2:

**Proposition 4.** *Let $p_0$ be any probability distribution, and let $Z = (Z_t)_{t \in [0,T]}$, $Z' = (Z'_t)_{t \in [0,T]}$ be two different processes satisfying*

$$dZ_t = b(Z_t, t)dt + \sigma(t)dB_t, \quad Z_0 \sim p_0,$$
$$dZ'_t = b'(Z'_t, t)dt + \sigma(t)dB_t, \quad Z'_0 \sim p_0.$$

*We define the distributions of $Z_t$ and $Z'_t$ as $p_t$ and $p'_t$, and the path measures of $Z$ and $Z'$ as $\mathbb{P}$ and $\mathbb{P}'$ respectively.*

*Suppose the following Novikov's condition:*

$$\mathbb{E}_{\mathbb{P}} \left[ \exp \left( \int_0^T \frac{1}{2} \int_x \sigma^{-2}(t) \|(b - b')(x, t)\|^2 dxdt \right) \right] < \infty. \tag{30}$$

*Then, the Radon-Nikodym derivative of $\mathbb{P}$ with respect to $\mathbb{P}'$ is*

$$\frac{d\mathbb{P}}{d\mathbb{P}'}(Z) = \exp \left\{ -\frac{1}{2} \int_0^T \sigma(t)^{-2} \|(b - b')(Z_t, t)\|^2 dt - \int_0^T \sigma(t)^{-1}(b - b')(Z_t, t)dB_t \right\},$$

*and therefore we have that*

$$KL(p_T \| p'_T) \leq KL(\mathbb{P} \| \mathbb{P}') = \int_0^T \frac{1}{2} \int_x p_t(x)\sigma(t)^{-2} \|(b - b')(x, t)\|^2 dxdt.$$

*Moreover, Chen et al. (2022) showed that if $\int_x p_t(x)\sigma^{-2}(t) \|(b - b')(x, t)\|^2 dx \leq C$ holds for some constant $C$ over all $t$, we have that*

$$KL(p_T \| p'_T) \leq \int_0^T \frac{1}{2} \int_x p_t(x)\sigma(t)^{-2} \|(b - b')(x, t)\|^2 dxdt,$$

*even if the Novikov's condition equation 30 is not satisfied.*

.

According to equation 27 and equation 29, we have

$$\int_x p_t(x)\beta(t)^{-1} \|\tilde{f}(\mathbf{X}, t) - \hat{f}(\mathbf{X}, t)\|dx \leq \frac{\beta(t)}{\sqrt{1 - e^{-\int_{t_0}^t \beta(s)ds}}} \delta \leq \sup_{t \in [t_0, T]} \frac{\beta(t)}{\sqrt{1 - e^{-\int_{t_0}^t \beta(s)ds}}} \delta, \tag{31}$$

$$\int_x p_t(x)\beta(t)^{-1} \|\tilde{f}(\mathbf{X}, t) - f(\mathbf{X}, t)\|dx \leq \frac{\beta(t)}{\sqrt{1 - e^{-\int_{t_0}^t \beta(s)ds}}} \delta \leq \sup_{t \in [t_0, T]} \frac{\beta(t)}{\sqrt{1 - e^{-\int_{t_0}^t \beta(s)ds}}} \delta. \tag{32}$$

Therefore, we can apply Proposition 4 to obtain upper bounds for the KL-divergences, which leads to

$$\mathrm{KL}(\tilde{p}_{t_0} | \hat{p}_{t_0}) \leq \int_{t_0}^T \frac{1}{2} \int_x p_t(x)\beta(t)^{-1} \|\tilde{f}(\mathbf{X}, t) - \hat{f}(\mathbf{X}, t)\|dx$$
$$\leq \delta \int_{t_0}^T \frac{1}{2} \frac{\beta(t)}{\sqrt{1 - e^{-\int_{t_0}^t \beta(s)ds}}} dt \tag{33}$$

and

$$\mathrm{KL}(\tilde{p}_{t_0} | p_{t_0}) \leq \int_{t_0}^T \frac{1}{2} \int_x p_t(x)\beta(t)^{-1} \|\tilde{f}(\mathbf{X}, t) - f(\mathbf{X}, t)\|dx$$
$$\leq \delta \int_{t_0}^T \frac{1}{2} \frac{\beta(t)}{\sqrt{1 - e^{-\int_{t_0}^t \beta(s)ds}}} dt. \tag{34}$$

$\square$

**Remark 1.** *Under the SDE formulation, the forward process terminates at a sufficiently large time T. Also, since the score functions blow up at $t \approx 0$, an early-stopping time $t_0$ is commonly adopted to avoid such issue (Song & Ermon (2020); Nichol & Dhariwal (2021)). When $t_0$ is sufficiently small, the distribution of $\mathbf{X}_{t_0}$ in the forward process is close enough to the real data distribution.*

# C    DETAILS OF CONDITIONING AND ALGORITHMS

## C.1    MATHEMATICAL FORMULATION OF TEXTURAL CONDITIONS IN SECTION 4.1

Denote a chord progression by $\mathcal{C}$, where $\mathcal{C}(l)$ denotes the chord at time $l \in [\![1, L]\!]$. Let $\gamma_{\mathcal{C}}(l) \subset [\![1, H]\!]$ denote the set of pitch index $h$ that belongs to the pitch classes included in chord $\mathcal{C}(l)$.[11], and let $\gamma_{\mathcal{R}} \subset [\![1, L]\!]$ denote the set of onset time indexes corresponding to rhythmic pattern $\mathcal{R}$. We define the following versions of representations for the condition:

- When harmonic ($\mathcal{C}$) and rhythmic ($\mathcal{R}$) conditions are both provided, the corresponding conditional piano roll $\boldsymbol{M}^{\text{cond}}(\mathcal{C}, \mathcal{R})$ is given element-wise by $M^{\text{cond}}{}_{lh}(\mathcal{C}, \mathcal{R}) = \mathbf{1}\{l \in \gamma_{\mathcal{R}}\}\mathbf{1}\{h \in \gamma_{\mathcal{C}}(l)\}$, meaning that the $(l, h)$-element is 1 if pitch index $h$ belongs to chord $\mathcal{C}(l)$ and there is onset notes at time $l$, and 0 otherwise.

- When only harmonic ($\mathcal{C}$) condition is provided, the corresponding piano roll $\boldsymbol{M}^{\text{cond}}(\mathcal{C})$ is given element-wise by $M^{\text{cond}}{}_{lh}(\mathcal{C}) = -1 - \mathbf{1}\{h \in \gamma_{\mathcal{C}}(l)\}$, meaning that the $(l, h)$-element is $-2$ if pitch index $h$ belongs to chord $\mathcal{C}(l)$, and $-1$ otherwise.

Figure 2 provides illustrative examples of $\boldsymbol{M}^{\text{cond}}(\mathcal{C}, \mathcal{R})$ and $\boldsymbol{M}^{\text{cond}}(\mathcal{C})$. The use of $-2$ and $-1$ (rather than 1 and 0) in the latter case ensures that the model can fully capture the distinctions between the two scenarios, as a unified model will be trained on both types of conditions.

## C.2    ADDITIONAL ALGORITHMS IN SECTION 4.2

In this section, we provide the following algorithm: fine-grained sampling guidance additionally with rhythmic regularization, fine-grained sampling guidance combined with DDIM sampling.

Let $\mathcal{B}$ denote the rhythmic regularization. Specifically, we have the following types of regularization:

- $\mathcal{B}_1$: Requiring exactly $N$ onset of a note at time position $l$, i.e., $\sum_{h \in [\![1, H]\!]} M_{lh} = N$

- $\mathcal{B}_2$: Requiring at least $N$ onsets at time position $l$, i.e.,

  $\exists \boldsymbol{h} \subset [\![1, H]\!]$, or $\exists \boldsymbol{h} \subset [\![1, H]\!] \backslash \omega_{\mathcal{K}}(l)$ if harmonic regularization is jointly included

  such that $M_{l\boldsymbol{h}} = 1$, and $|\boldsymbol{h}| \geq N$

- $\mathcal{B}_3$: Requiring no onset of notes at time position $l$, i.e., $\forall h \in [\![1, H]\!]$, $M_{lh} = 0$

Let the set of $\boldsymbol{M}$ satisfying a specific regularization $\mathcal{B}$ be denoted as $\mathbb{M}_{\mathcal{B}}$, and the corresponding set of $\boldsymbol{X}$ be denoted as $\tilde{\mathbb{M}}_{\mathcal{B}}$, note that this includes the case where multiple requirements are satisfied, resulting in

$$\tilde{\mathbb{M}}_{\mathcal{B}} = \tilde{\mathbb{M}}_{\mathcal{B}_1, \mathcal{B}_2, \dots} = \tilde{\mathbb{M}}_{\mathcal{B}_1} \cap \tilde{\mathbb{M}}_{\mathcal{B}_2} \cap \dots .$$

The correction of predicted noise score is then formulated as

$$\tilde{\varepsilon}_{\theta}(\boldsymbol{X}_t, t | \mathcal{C}, \mathcal{R}) = \arg\min_{\varepsilon} \quad \|\varepsilon - \widehat{\varepsilon}_{\theta}(\boldsymbol{X}_t, t | \mathcal{C}, \mathcal{R})\|$$

$$\text{s.t.} \quad \left(\frac{\boldsymbol{X}_t - \sqrt{1 - \bar{\alpha}_t}\varepsilon}{\sqrt{\bar{\alpha}_t}}\right) \in \tilde{\mathbb{M}}_{\mathcal{B}}. \tag{35}$$

Further, we can perform predicted noise score correction with joint regularization on rhythm and harmony, resulting in the corrected noise score

$$\tilde{\varepsilon}_{\theta}(\boldsymbol{X}_t, t | \mathcal{C}, \mathcal{R}) = \arg\min_{\varepsilon} \quad \|\varepsilon - \widehat{\varepsilon}_{\theta}(\boldsymbol{X}_t, t | \mathcal{C}, \mathcal{R})\|$$

$$\text{s.t.} \quad \left(\frac{\boldsymbol{X}_t - \sqrt{1 - \bar{\alpha}_t}\varepsilon}{\sqrt{\bar{\alpha}_t}}\right) \in (\mathbb{R}^{L \times H} \backslash \mathbb{W}'_{\mathcal{K}}) \cap \tilde{\mathbb{M}}_{\mathcal{B}}. \tag{36}$$

---

[11]For example, when $\mathcal{C}(l) = $ C major (consisting of pitch classes C, E and G), $\gamma_{\mathcal{C}}$ includes all pitch values corresponding to the three pitch classes across all octaves.

We for example provide a element-wise solution of $\tilde{\varepsilon}_\theta(\boldsymbol{X}_t, t|\mathcal{C}, \mathcal{R})$ defined by problem (35). For given $l$, suppose $\mathcal{B}(l)$ takes the form of $\mathcal{B}_2$, for simplicity take $N = 1$. This gives $\tilde{\varepsilon}_{\theta,lh} = \hat{\varepsilon}_{\theta,lh}$ if $\max_h \mathbb{E}[\mathbf{X}_0|\boldsymbol{X}_t]_{hl} \geq \frac{1}{2}$ and $\mathbb{E}[\mathbf{X}_0|\boldsymbol{X}_t]_{hl} = \frac{1}{2}$, $h = \arg\max_h \mathbb{E}[\mathbf{X}_0|\boldsymbol{X}_t]_{hl}$, i.e.,

$$\tilde{\varepsilon}_{\theta,lh} = \frac{1}{\sqrt{1 - \bar{\alpha}_t}} \left( X_{t,lh} - \frac{\sqrt{\bar{\alpha}_t}}{2} \right),$$

if $\max_h \mathbb{E}[\mathbf{X}_0|\boldsymbol{X}_t]_{hl} < \frac{1}{2}$. The correction applied to predicted $\mathbf{X}_0$ ($\mathbb{E}[\mathbf{X}_0|\boldsymbol{X}_t]$) is illustrated in the following figure 4.

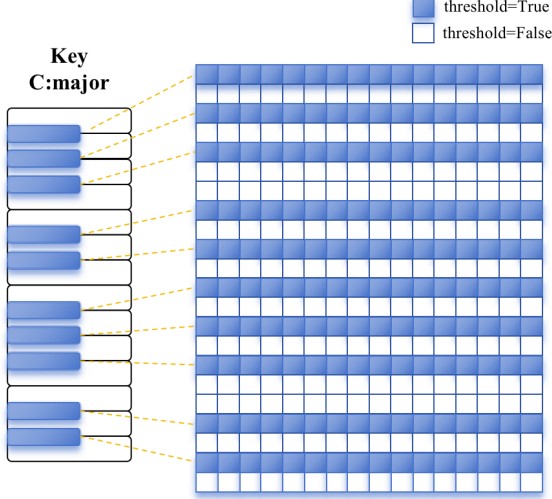

(a) Fine-grained control for $\mathbb{E}[\mathbf{X}_0|\boldsymbol{X}_t] \in \mathbb{R}^{L \times H} \backslash \mathbb{W}'_\mathcal{K}$. The colored spots denote places that we require $\mathbb{E}[\mathbf{X}_0|\boldsymbol{X}_t]_{lh} \leq \frac{1}{2}$.

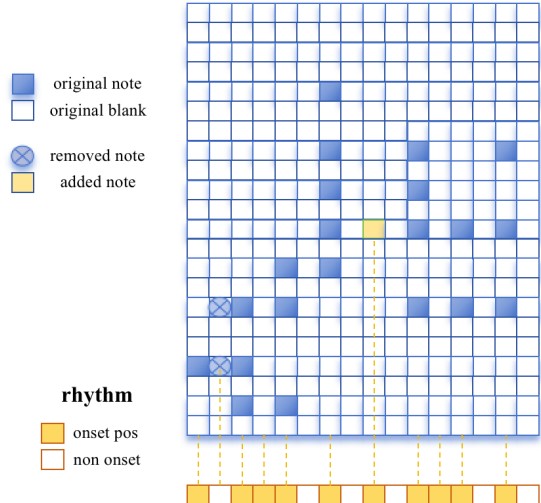

(b) Fine-grained control for $\mathbb{E}[\mathbf{X}_0|\boldsymbol{X}_t] \in \mathbb{W}'_\mathcal{B}$. Original notes are removed at $l$ if $\mathcal{B}_3$ is applied. Otherwise if $\mathcal{B}_1$ is applied and currently no note exists, the "most likely notes" (i.e., at $h = \arg\max \mathbb{E}[\mathbf{X}_0|\boldsymbol{X}_t]_{lh}$) are added.

Figure 4: Illustration of fine-grained control on predicted $\mathbf{X}_0$.

---

**Algorithm 2:** DDPM sampling with fine-grained textural guidance

---

**Input:** Input parameters: forward process variances $\beta_t$, $\bar{\alpha}_t = \prod_{s=1}^t \beta_t$, backward noise scale
$\sigma_t$, chord condition $\mathcal{C}$, key signature $\mathcal{K}$, rhythmic condition $\mathcal{R}$, rhythmic guidance $\mathcal{B}$

**Output:** generated piano roll $\tilde{\mathbf{M}} \in \{0,1\}^{L \times H}$

**1** $\mathbf{X}_T \sim \mathcal{N}(0, \boldsymbol{I})$;

**2 for** $t = T, T-1, \ldots, 1$ **do**

**3**     Compute guided noise prediction $\widehat{\varepsilon}_\theta(\boldsymbol{X}_t, t | \mathcal{C}, \mathcal{R})$;

**4**     Perform noise correction: derive $\tilde{\varepsilon}_\theta(\boldsymbol{X}_t, t | \mathcal{C}, \mathcal{R})$ optimization equation 36;

**5**     Compute $\tilde{\mathbf{X}}_{t-1}$ by plugging the corrected noise $\tilde{\varepsilon}_\theta(\boldsymbol{X}_t, t | \mathcal{C}, \mathcal{R})$ into equation 2

**6 end**

**7** Convert $\tilde{\mathbf{X}}_0$ into piano roll $\tilde{\mathbf{M}}$

**8 return** $output$;

---

We additionally remark that the fine-grained sampling guidance is empirically effective with the DDIM sampling scheme, which drastically improves the generation speed. Specifically, select subset $\{\tau_i\}_{i=1}^m \subset [\![1, T]\!]$, and denote

$$\mathbf{X}_{\tau_{i-1}} = \sqrt{\bar{\alpha}_{\tau_{i-1}}} \left( \frac{\mathbf{X}_t - \sqrt{1 - \bar{\alpha}_{\tau_i}} \widehat{\varepsilon}_\theta(\mathbf{X}_{\tau_i}, \tau_i)}{\sqrt{\bar{\alpha}_{\tau_i}}} \right) + \sqrt{1 - \bar{\alpha}_{\tau_{i-1}} - \sigma_{\tau_i}^2} \widehat{\varepsilon}_\theta(\mathbf{X}_{\tau_i}, \tau_i) + \sigma_{\tau_i} \varepsilon_{\tau_i},$$

we similarly perform the DDIM noise correction

$$\tilde{\varepsilon}_\theta(\boldsymbol{X}_{\tau_i}, \tau_i | \mathcal{C}, \mathcal{R}) = \arg\min_{\boldsymbol{\varepsilon}} \quad \|\boldsymbol{\varepsilon} - \widehat{\varepsilon}_\theta(\boldsymbol{X}_{\tau_i}, \tau_i | \mathcal{C}, \mathcal{R})\|$$

$$\text{s.t.} \quad \left( \frac{\boldsymbol{X}_t - \sqrt{1 - \bar{\alpha}_{\tau_i}} \boldsymbol{\varepsilon}}{\sqrt{\bar{\alpha}_{\tau_i}}} \right) \in (\mathbb{R}^{L \times H} \backslash \mathbb{W}'_{\mathcal{K}}) \cap \tilde{\mathbb{M}}_{\mathcal{B}}.$$

on each step $i$.

## D  COMPARISON WITH RELATED WORKS

We provide a detailed comparison between our method and two related works in controlled diffusion models with constrained or guided intermediate sampling steps:

**Comparison with reflected diffusion models** In Lou & Ermon (2023), a bounded setting is used for both the forward and backward processes, ensuring that the bound applies to the training objective as well as the entire sampling process. In contrast, we do not adopt the framework of bounded Brownian motion, because we do not require the entire sampling process to be bounded within a given domain; instead, we only enforce that the final sample outcome aligns with the constraint. While Lou & Ermon (2023) enforces thresholding on $\mathbf{X}_t$ in both forward and backward processes, our approach is to perform a thresholding-like projection method on the predicted noise $\varepsilon_\theta(\mathbf{X}_t, t)$, interpreted as noise correction.

**Comparison with non-differentiable rule guided diffusion** Huang et al. (2024) guides the output with musical rules by sampling multiple times at intermediate steps, and continuing with the sample that best fits the musical rule, producing high-quality, rule-guided music. Our work centers on a different aspect, prioritizing precise control to tackle the challenges of accuracy and regularization in symbolic music generation. Also, we place additional emphasis on sampling speed, ensuring stable generation of samples within seconds to facilitate interactive music creation and improvisation.

## E  NUMERICAL EXPERIMENT DETAILS

### E.1  DETAILED DATA REPRESENTATION

The two-channel version of piano roll with with both harmonic and rhythm conditions ($\mathbf{M}^{\text{cond}}(\mathcal{C}, \mathcal{R})$) and with harmonic condition ($\mathbf{M}^{\text{cond}}(\mathcal{C})$) with onset and sustain are represented as:

- $\mathbf{M}^{\text{cond}}(\mathcal{C}, \mathcal{R})$: In the first channel, the $(l, h)$-element is 1 if there are onset notes at time $l$ and pitch index $h$ belongs to the chord $\mathcal{C}(l)$, and 0 otherwise. In the second channel, the

$(l, h)$-element is 1 if pitch index $h$ belongs to the chord $\mathcal{C}(l)$ and there is no onset note at time $l$.

- $\mathbf{M}^{\text{cond}}(\mathcal{C})$: In both channels, the $(l, h)$-element is 1 if pitch index $h$ belongs to the chord $\mathcal{C}(l)$, and 0 otherwise.

In each diffusion step $t$, the model input is a concatenated 4-channel piano roll with shape $4 \times L \times 128$, where the first two channels correspond to the noisy target $\mathbf{X}_t$ and the last two channels correspond to the condition $\mathbf{M}^{\text{cond}}$ (either $\mathbf{M}^{\text{cond}}(\mathcal{C}, \mathcal{R})$ or $\mathbf{M}^{\text{cond}}(\mathcal{C})$). The output is the noise prediction $\hat{\varepsilon}_\theta$, which is a 2-channel piano roll with the same shape as $\mathbf{X}_t$. For the accompaniment generation experiments, we provide melody as an additional condition, which is also represented by a 2-channel piano roll with shape $2 \times L \times 128$, with the same resolution and length as $\mathbf{X}$. The melody condition is also concatenated with $\mathbf{X}_t$ and $\mathbf{M}^{\text{cond}}$ as model input, which results in a full 6-channel matrix with shape $6 \times L \times 128$.

### E.2 TRAINING AND SAMPLING DETAILS

We set diffusion timesteps $T = 1000$ with $\beta_0 = 8.5e{-}4$ and $\beta_T = 1.2e{-}2$. We use AdamW optimizer with a learning rate of $5e{-}5$, $\beta_1 = 0.9$, and $\beta_2 = 0.999$. We train for 20 epochs with batch size 16, resulting in 985 steps in each epoch.

To speed up the sampling process, we select a sub-sequence of length 10 from $\{1, \cdots, T\}$ and apply the accelerated sampling process in Song et al. (2020a). It takes 0.4 seconds to generate the 4-measure accompaniment on a NVIDIA RTX 6000 Ada Generation GPU.

### E.3 EXPERIMENTS ON SYMBOLIC MUSIC GENERATION GIVEN ONLY CHORD CONDITIONS

As mentioned in Section 5.1, we also run numerical experiments on symbolic music generation tasks given only chord condition. However, compared with the accompaniment generation task, we remark that this experiment does not have enough effective basis for comparison.

For the accompaniment generation task, we evaluate the cosine similarity of chord progression between the generated samples and the ground truth, as well as the macro overlapping area (MOA) of features including note pitch, duration, and note density. The comparison with ground truth on those features make sense in the accompaniment generation task, because the leading melody inherently contains many constraints on the rhythm and pitch range of the accompaniment, ensuring coherence with the melody. Thus, similarity with ground truth on those metrics serves as an indicator of how well the generated samples adhere to the melody.

However, in symbolic music generation conditioned only on a chord sequence, while chord progression similarity remains comparable (as the chord sequence is provided), evaluating MOA features against ground truth is less informative. This is because multiple different pitch range and rhythm could appropriately align with a given chord progression, making deviations from the ground truth in these features less indicative of sample quality. Therefore, chord similarity emerges as the sole applicable metric in this context.

Additionally, WholeSongGen's architecture does not support music generation conditioned solely on chord progressions, as it utilizes a shared piano-roll for both chord and melody, rendering it unsuitable for comparison. Conversely, GETMusic facilitates the generation of both melody and piano accompaniment based on chord conditions, allowing for a viable comparison.

Consequently, we present results focusing on chord similarity between our model and GETMusic. For our model, we evaluate performance under two conditions: with both conditioning and control during training and sampling, and with conditioning during training but without control during sampling. The outcomes, summarized in Table 4, indicate that our fully controlled FTG method surpasses both the one without sampling control and GETMusic.

## F SUBJECTIVE EVALUATION

To compare performance of our FTG method against the baselines (WholeSongGen and GETMusic), we prepared 6 sets of generated samples, with each set containing the melody paired with

| Methods | FTG (Ours) | FTG, only Training control | GETMusic |
|---|---|---|---|
| **Chord Similarity** | **$0.676 \pm 0.007$** | $0.645 \pm 0.008$ | $0.499 \pm 0.013$ |

Table 4: Evaluation of the similarity with ground truth, chord-conditioned music generation.

accompaniments generated by FTG, WholeSongGen, and GETMusic, along with the ground truth accompaniment. This yields a total of $6 \times 4 = 24$ samples. The samples are presented in a randomized order, and their sources are not disclosed to participants. Experienced listeners assess the quality of samples in 5 dimensions: creativity, harmony (whether the accompaniment is in harmony with the melody), melodiousness, naturalness and richness, together with an overall assessment.

### F.1 BACKGROUND OF PARTICIPANTS

To evaluate the musical background of the participants, we first present the following questions:

- How many instruments (including vocal) are you playing or have you played?

- Please list all instruments (including vocal) that you are playing or have played.

- What is the instrument (including vocal) you have played the longest, and how many years have you been playing it? (e.g., piano, 3 years)

We recruited 31 participants with substantial musical experience for our survey. The number of instruments these participants play range from 0 to 5, with an average value of 2.03, and a standard deviation of 1.31. Examples of instrument played include piano, violin, vocal, guitar, saxphone, Dizi, Yangqin and Guzheng. The average years of playing has an average of 8.61 and standard deviation of 8.08. Specifically, the percentage of participants with $\geq 3$ years of playing music is 67.74%, and the percentage of participants with $\geq 10$ years of playing music is 45.16%. The distributions are given in the following figure 5.

### F.2 EVALUATION QUESTIONS

Thank you for taking the time to participate in this experiment. You will be presented with 6 sets of clips, each containing 4 clips. The first clip in each set features the melody alone, while the remaining three include the melody accompanied by different accompaniments. After listening to each clip, please evaluate the accompaniments in the following dimensions based on your own experience.

- Does the accompaniment sound pleasant to you?

- How would you rate the richness of the accompaniment?

- Does the accompaniment sound natural?

- Does the accompaniment align well with the melody?

- Does the accompaniment demonstrate creativity?

- Please give an overall score for the clip.

For each question, participants are provided with a Likert scale ranging from 1 to 5, where 1 represents "very poor" and 5 represents "very good."

## G  REPRESENTATIVE EXAMPLES OF SAMPLING CONTROL

In this section, we provide empirical examples of how model output is reshaped by fine-grained correction in Figure 6. Notably, harmonic control not only helps the model eliminate incorrect notes, but also guides it to replace them with correct ones.

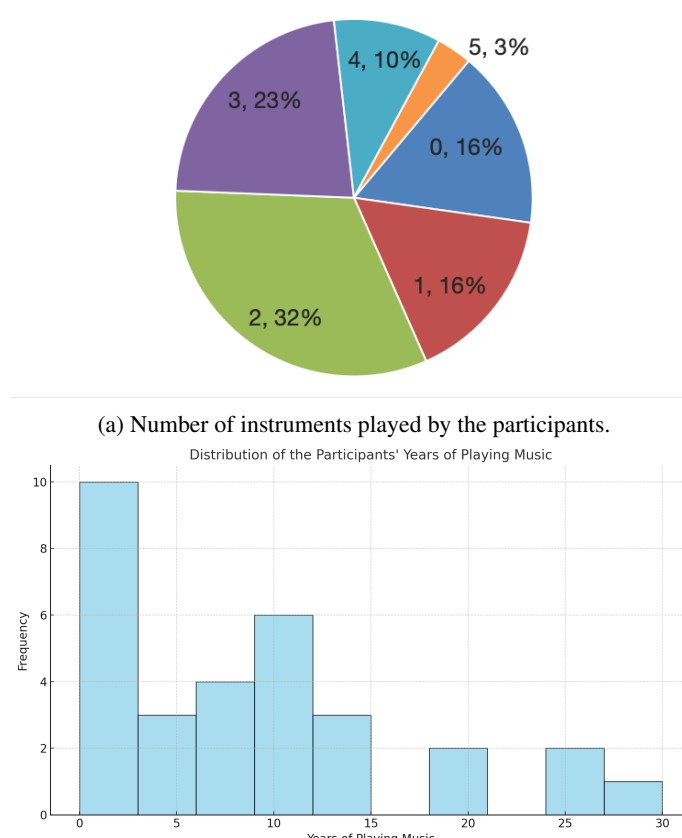

(a) Number of instruments played by the participants.

(b) Distribution of the participants' years of playing instruments.

Figure 5: Information of the musical background of the participants in the subjective evaluation.

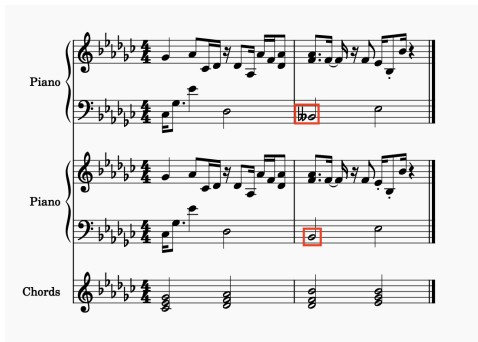
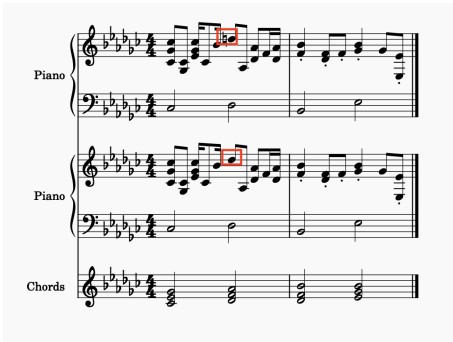

(a) An example of replacing an out-of-key note B♭♭ with the in-key note B♭.

(b) An example of replacing an out-of-key note D♮ with the in-key note D♭.

Figure 6: Examples resulting from symbolic music generation with FTG. The first track is generated without key-signature control in sampling, the second track is generated with key-signature sampling control. The third track presents the chord condition. In each subfigure, the tracks are generated with the same conditions and the same set of noise.

## H  THE EFFECT OF GUIDANCE WEIGHT FOR CLASSIFIER-FREE GUIDANCE

In Section 4.1, we discussed the implementation of classifier-free guidance for rhythmic patterns, designed to enable the model to generate outputs under varying levels of conditioning. Specifically,

we randomly apply conditions with or without rhythmic patter in the process of training. This approach ensures that the model can function effectively with both chord and rhythmic conditions or with chord conditions alone. Following Ho & Salimans (2022), when generating with both chord and rhythmic conditions, the guided noise prediction at timestep $t$ is computed as:

$$
\begin{aligned}
\varepsilon_\theta(\mathbf{X}_t, t | \mathcal{C}, \mathcal{R}) =& \varepsilon_\theta(\mathbf{X}_t, \boldsymbol{M}^{\mathrm{cond}}(\mathcal{C}), t) \\
&+ w \cdot \left[ \varepsilon_\theta(\mathbf{X}_t, \boldsymbol{M}^{\mathrm{cond}}(\mathcal{C}, \mathcal{R}), t) - \varepsilon_\theta(\mathbf{X}_t, \boldsymbol{M}^{\mathrm{cond}}(\mathcal{C}), t) \right],
\end{aligned}
$$

where $\varepsilon_\theta(\mathbf{X}_t, \boldsymbol{M}^{\mathrm{cond}}(\mathcal{C}, \mathcal{R}), t)$ is the model's predicted noise without rhythmic condition, and $\varepsilon_\theta(\mathbf{X}_t, \boldsymbol{M}^{\mathrm{cond}}(\mathcal{C}, \mathcal{R}), t)$ is the model's predicted noise with rhythmic condition, and $w$ is the guidance weight.

The literature has consistently demonstrated that the guidance weight $w$ plays a pivotal role in balancing diversity and stability in generation tasks (Ho & Salimans, 2022; Chang et al., 2023; Gao et al., 2023; Lin et al., 2024). In general, a lower weight $w$ enhances sample diversity and quality, but this may come at the cost of deviation from the provided conditions. Conversely, higher values of $w$ promote closer adherence to the conditioning input, but excessively high $w$ can degrade output quality by over-constraining the model, resulting in less natural or lower-quality samples.

In this section, we hope to investigate the effect of the guidance weight $w$ on our music generation task. We focus on the same accompaniment generation task as mentioned in Section 5. To measure the samples' adherence to rhythmic controls, we use the rhythm of the ground truth as the rhythmic condition and assess the overlapping area (OA) of note duration and note density between the generated and ground-truth samples. Additionally, we measured the percentage of out-of-key notes as a proxy for sample quality. In these experiments, we only use the fine-grained control in training, but do not insert any sampling control so that we can evaluate the inherent performance of the models themselves. The experiments were conducted across a range of guidance weights ($w$ from $0.5$ to $10$), and he results are summarized in Table 5.

| Values of $w$ | % Out-of-Key Notes | OA (duration) | OA (note density) |
|:---:|:---:|:---:|:---:|
| 0.5 | 1.3% | 0.592 ±0.005 | 0.803 ±0.004 |
| 1.0 | 1.4% | 0.617 ±0.005 | 0.830 ±0.003 |
| 3.0 | 1.7% | 0.644 ±0.003 | 0.848 ±0.003 |
| 5.0 | 2.6% | 0.638 ±0.005 | 0.846 ±0.003 |
| 7.5 | 6.0% | 0.643 ±0.005 | 0.829 ±0.004 |
| 10.0 | 14.3% | 0.630 ±0.005 | 0.779 ±0.005 |

Table 5: Comparison of the results with and without control in the sampling process.

The findings indicate that as the guidance weight $w$ increases, the percentage of out-of-key notes rises, suggesting that lower $w$ values yield higher-quality samples. Meanwhile, the OA of duration and note density improves as $w$ increases from $0.5$ to $3.0$, indicating better alignment with rhythmic conditions. However, when $w$ exceeds $5.0$, a notable decline is observed in both the OA metrics and the percentage of out-of-key notes. This degradation is likely due to a significant drop in sample quality at excessively high $w$ values, where unnatural outputs undermine adherence to the rhythmic conditions. These observations are coherent with the existing results about the trade-off between sample quality and adherence to conditions in literature.

## I  DISCUSSION

The role of generative AI in music and art remains an intriguing question. While AI has demonstrated remarkable performance in fields such as image generation and language processing, these domains possess two characteristics that symbolic music lacks: an abundance of training data and well-designed objective metrics for evaluating quality. In contrast, for music, it is even unclear whether it is necessary to set the goal as generating compositions that closely resemble[12] some "ground truth".

In this work, we apply fine-grained sampling control to eliminate out-of-key notes, ensuring that generated music adheres to the most common harmonies and chromatic progressions. This approach allows the model to consistently and efficiently produce music that is (in some ways) "pleasing to the ear". While suitable for the task of quickly creating large amounts of mediocre pieces, such models have a limited capability of replicating the artistry of a real composer, of creating sparkles with unexpected "wrong" keys by themselves.

---

[12]or, in what sense?

