# OpenReview forum: "Symbolic Music Generation with Fine-grained Interactive Textural Guidance"
_ICLR.cc/2025/Conference — Submitted to ICLR 2025_

### Official Review · Reviewer_aY42 · 2024-10-28

**Soundness:** 2
**Presentation:** 1
**Contribution:** 2
**Rating:** 3
**Confidence:** 4

**Summary:**

This paper addresses challenges in symbolic music generation, specifically in harmonic precision and rhythmic regularity. The authors present a Fine-grained Textural Guidance (FTG) framework for symbolic music generation within a diffusion model. The proposed method applies harmonic and rhythmic guidance during both training and sampling, incorporating classifier-free guidance and noise correction mechanisms tailored to piano roll representations, aiming to improve accuracy and alignment in generated symbolic music. Experimental results on the POP909 dataset show notable improvements over baseline models in chord similarity and feature distribution alignment with ground truth. A demo page with interactive posibilitied further demonstrates the model’s performance in real-time generation.

**Strengths:**

The paper presents an innovative approach to symbolic music generation by incorporating Fine-grained Textural Guidance (FTG) within a diffusion model, addressing challenges in harmonic precision and rhythmic regularity. The originality primarily arises from the application of existing diffusion techniques, like classifier-free guidance and noise correction, to symbolic music data in a way that addresses specific musical challenges, such as harmonic precision and rhythmic regularity. The promising results were demonstrated on the POP909 dataset. The improvements on the metrics of chord similarity and feature distribution are strong. Overall, the idea of FTG contributing for advancing precision in symbolic music generation is interesting. The demo page is good and samples sound adequate.

**Weaknesses:**

Unclear paper objectives:

The paper emphasizes the improvement of "wrong" (out-of-key) notes and improvements in rhythmic regularity, but it remains unclear how directly these challenges are addressed in the model's performance. As I understand it, "wrong" notes are entirely removed—a choice that is itself debatable—but what about rhythmic coherence? It may not always be desirable to remove "wrong" notes altogether; using them within the “right” context could result in more musically expressive outcomes (this was specifically discussed in the paper but the “wrong” notes were just removed). The strategy raises the question of whether such strong conditioning is ideal. Machine learning models might benefit from the ability to generate “wrong” notes intentionally, given the appropriate context. For rhythm, this conditioning seems more justified, as it acts as guidance rather than a strict restriction. additonally, while metrics used in the paper show improvement, it’s not clear if these gains specifically reflect solutions to the highlighted challenges. Direct metrics for these aspects of “wrong” notes and  rhythmic regularity would be helpful following such an introduction.

The scope of the method’s effectiveness for music generation versus arrangement generation could also be clarified. From what I see, the only task addressed here is accompaniment generation based on melody and chord progression. This task is also not fully clear. Normally in the similar paper they define accompaniment generation based on the melody and melody generation based on the accompaniments as separate task. The task itself I suppose Is not defined and communicated clearly. If generating from scratch isn’t feasible, that detail is important and might be better reflected in the paper’s title. Rather than "symbolic music generation," a more accurate title might be "symbolic accompaniment generation."


Theoretical claims on out-of-key notes:

In the problem statement section, from line 159 onward and Appendix C1 starting at line 742, the authors attempt to show how generative models struggle with out-of-key notes due to statistical limitations. While the intention is clear, the approach relies heavily on complex mathematical properties and assumptions, which at times give the impression of “formula-stuffing” without fully substantiating the claim. As someone not deeply specialized in this particular statistical approach, I find the reliance on restrictive assumptions, such as Properties I and II, and the emphasis on Total Variation distance in high dimensions, to be somewhat disconnected from the practical musical context. Similarly, the reference to the curse of dimensionality and other high-dimensional challenges may not directly translate to discrete symbolic music data. The original work by Fu et al. (2024) that inspired this seems more suited for continuous data, raising questions about its direct application here. Additionally, because music is inherently subjective and cultural, definitions of consonance and dissonance vary. The paper’s categorization of “good” versus “bad” contexts for out-of-key notes appears to lack a consideration of genre or historical period, which can heavily influence musical perception. This is a thoughtful attempt to formalize the problem, but it may not fully capture the nuances of musical interpretation.

in the end i don'tundestand why this took so much space in the paper if the plan was just to remove those "wrong" notes alltogether? I'm a little confused here.

Limitations of using 4/4 meter exclusively:

Another point to consider is the exclusion of triple meter pieces. This decision restricts the dataset to pieces in 4/4 (or derivative meters), which might simplify the model’s task and consequently limit the diversity and complexity of the generated output. Further explanation of this choice would be helpful.

Evaluation metrics:

The authors introduce their own evaluation metrics, which align well with the paper’s goals, but it is unclear how well these metrics relate to human perception. While the paper includes quantitative evaluations of chord similarity and overlapping area (OA) for various musical features, additional discussion on perceptual evaluations, such as subjective quality assessments, could offer insight into how these improvements translate to listener experience.

No Classifier-free guidance effect shown in results:

On the line 325, the use of classifier-free guidance is mentioned, but the results lack clarity on how it was implemented and whether it effectively enhanced variation in generation. While the fine-grained guidance is explained clearly, further elaboration on classifier-free guidance and its parameterization would improve understanding. A comparative analysis of FTG with and without noise correction (e.g., results for models using only harmonic guidance) could also highlight the contributions of individual components.

absence of a conclusion section:

Another notable aspect is the absence of a conclusion section, which, though possibly intentional, lends the paper an abrupt ending, leaving certain results and implications under-discussed.

**Questions:**

Questions and Suggestions for Improvement

From the comments above here are some questions/suggestions for the authors:

1. Direct Metrics for Rhythmic Coherence and “Wrong” Note Removal:

Could you provide specific metrics or examples that demonstrate how the method improves rhythmic coherence? We see the metrics improved but does this mean rhythmic coherence improved? If yese how? Some discussion on this would be very helpful.
Additionally, discussing potential trade-offs of completely removing "wrong" (out-of-key) notes versus allowing some flexibility would help clarify the model's approach to variability and musical adaptability. Some form of human evaluation might be helpful here, as I can’t think of an objective metric that would adequately measure this. Music often relies on a balance of anticipation and surprise, so eliminating “wrong” notes entirely could risk making the output less interesting—though this might be acceptable if the AI system is intended as a simple compositional tool. Regardless, it would be beneficial to reflect this consideration somehow in the results section.

2. Clear Task Definition for Accompaniment versus Music Generation:

Could you please more explicitly define the specific music generation task(s) you are addressing in this paper? So, it basically generates both melody or chords on the bases of both? This is a little unclear. Is this a new task? If so then how do we compare this to baselines where it is clearly separate tasks of melody from chords and cords form melody. Adding a clear statement on the task definition and scope in the introduction would enhance clarity. Also discussing how this corresponds to baselines.

3. Intuitive Explanation of Theoretical Analysis:

The theoretical claims on out-of-key notes ( from line 159 onward and Appendix C1 starting at line 742) rely heavily on complex mathematical assumptions, which for me felt somewhat disconnected from practical music generation. Could you provide a more intuitive explanation of how this theoretical analysis relates to practical music generation challenges? What does this analysis really say? Additionally, a discussion on the limitations of applying this statistical framework (that was developed for continuous data) to discrete symbolic music data would add valuable context.

4. Rationale for Excluding Triple Meter Pieces:

The exclusion of triple meter pieces limits the dataset to pieces in 4/4 (or derivative meters). Could you clarify the rationale for this exclusion and discuss how it might impact the generalizability of the results?

5. Results on Classifier-Free Guidance Implementation:

An ablation study or comparative results showing the impact of classifier-free guidance on output diversity would also enhance understanding.

6. Request for a Conclusion Section:

Adding a conclusion summarizing the key findings, discussing limitations, and proposing future work directions would improve the structure and provide a sense of closure.

**Details Of Ethics Concerns:**

While I appreciate the authors’ efforts to address reviewer concerns, the sudden improvements in both quantitative and subjective evaluations appear reactive rather than reflective of a carefully planned methodology. This creates a perception of selective reporting to present their method in a more favorable light.

I am flagging this submission for an ethics review due to serious concerns about the transparency and validity of the results presented during the rebuttal process. Specifically:

1. **Sudden Improvements in Table 1**:
   The authors have significantly revised Table 1, claiming improved metrics due to increasing the number of training epochs from 10 (in the original submission) to 20 (during rebuttal). This change fundamentally alters the experimental setup described in the initial submission, which raises the concerns. I don't undestand why wasn't 20 epochs training possible before? Retraining during the rebuttal period compromises the fairness of the review process, as the revised results were not derived under the same conditions as the original submission. While retraining might justify improved metrics, such updates should be presented as supplementary results rather than replacing the originally reported findings. This is not a good practive in my view.

2. **Introduction of a Subjective User Study**:
   The authors added a subjective evaluation section (Section 5.1.4) late in the rebuttal process, which raises questions about the timing and rigor of this user study.
   - When was this user study conducted?
   - How were participants recruited, and was the study sufficiently rigorous to justify the conclusions drawn?
   - Adding subjective evaluations during rebuttals feels like a rushed response rather than a deliberate component of the original methodology.

3. **Transparency and Reproducibility**:
   The lack of upfront communication about these substantial changes and the way the authors revised key results without clear distinction from the original submission undermines confidence in the reproducibility and integrity of the work.

This all raises in me a concern, and I believe this paper should not be approved without a serious inspection of the code, its reproducibility, and a thorough investigation of the user study details and correctness.

---

> ### Author Response · Authors · 2024-11-26
> **Response to Reviewer aY42**
>
> We appreciate the reviewer’s valuable suggestions. We have modified our manuscript and provide the following response to the reviewer’s questions, followed by some other comments.
>
> 1. Direct Metrics for Rhythmic Coherence and “Wrong” Note Removal:
>
> We provide an interface that can insert specific rhythmic control in part 3  (interactive real time generation) of our demo page (https://huggingface.co/spaces/interactive-symbolic-music/InteractiveSymbolicMusicDemo), under the “use customized chord” icon.
>
> We thank the reviewer for pointing out the potential trade-off in eliminating the wrong notes in harming the model’s ability to provide “surprises”. We have the following remarks on this aspect.
>
> (1) While the out-of-key notes are never necessarily “wrong”, it is observed that generative models often fail to create an accommodating context for such “surprises”. We have uploaded examples to part 1 of the demo page to demonstrate the phenomenon. Sometimes the generated accompaniment without control can be completely chaotic, completely mismatching the style of the melody  (e.g., example 1 and 3 in part 1 of our demo page https://huggingface.co/spaces/interactive-symbolic-music/InteractiveSymbolicMusicDemo). We also aim to explain this phenomenon using section 3.1, which now has been revised to be more straightforward.
>
> (2) Instead of eliminating a fixed note, a more important emphasis of our method is the precise control over the generation process,  aiming to avoid producing any "unwanted notes." Since the sampling control can be designed by the user of FTG, the appearance of "wrong notes" can also be intentionally incorporated as part of the sampling guidance rules. This method may also be used to provide precise control to generate a class of pre-specified music types.
>
> 2. Clear Task Definition for Accompaniment versus Music Generation
>
> The conditional guidance and sampling control proposed in our work can be integrated into existing diffusion-based frameworks for symbolic music generation, e.g., WholeSongGen[1]. Therefore, our methodology of applying precise control over generation is not limited to a specific music generation or accompaniment generation task. To avoid misunderstandings, we have changed the title of our work to “Efficient Fine-Grained Sampling Guidance for Diffusion-Based Symbolic Music Generation”, and we accordingly modified the structure of the content.
>
> 3. Intuitive Explanation of Theoretical Analysis
>
> We have revised the section by removing unnecessary modeling on key signatures. Our theoretical analysis aims to answer the following question: Why is it hard for models to eliminate “wrong notes” all by themselves via training? Our theoretical analysis points out that the difficulty comes from the requirement of precisely generating every pixel of the piano roll canvas, combined with a high data dimension. Admittedly, providing rigorous theoretical analysis often requires complicated modeling and assumptions. In our new version, we removed unnecessary notations and assumptions. We also supplement the theoretical analysis with empirical samples generated by models, where the out-of-key notes result in significant dissonance, as well as a numerical experiment calculating the percentage of such notes, if no sampling control in applied.
>
> 4. Rationale for Excluding Triple Meter Pieces
>
> Our method of fine-grained textural guidance can be applied to triple meter pieces as well as 4/4 pieces. The reason that we did not do experiments on triple meter pieces is lack of high-quality training data, as suggested in [1]. (“The capability of 3/4 song generation is limited, since the proportion of 3/4 songs in the dataset is pretty low.”)
>
> **References**
>
> [1] Ziyu Wang, Lejun Min, and Gus Xia. Whole-song hierarchical generation of symbolic music using cascaded diffusion models. The Twelfth International Conference on Learning Representations, 2024.

---

> ### Author Response · Authors · 2024-11-26
>
> 5. Results on Classifier-Free Guidance Implementation
>
> Classifier-free guidance enables a single model to produce both conditional and unconditional outputs seamlessly by interpolating between guided and unguided sampling, offering flexibility in use cases without requiring multiple models. Since providing rhythmic condition specifies the rhythmic pattern of the model output, whereas no rhythmic condition allows the model to generate any kind of rhythmic pattern, one can choose to not insert rhythmic condition so that the model can generate diversified rhythms. That is what we meant by saying that classifier-free guidance enable the model to generate under varying levels of conditioning.
>
> Thus, we do not intend to compare the results with and without using classifier-free guidance because that is not the focus of our proposed method. The main reason why we use classifier-free guidance is to obtain a unified model that can be both conditional on rhythmic patterns or unconditioned on rhythmic patters.
>
> We admit that the previous sentence "While conditional generation enhances output stability, it comes at the cost of reduced sample variation, particularly when rhythmic conditions are applied." might be confusing, and we have removed it in the updated manuscript.
>
> 6. Request for a Conclusion Section
>
> We thank the reviewer for the suggestion. We have added a conclusion section in our manuscript.
>
> 7. Comments on the evaluation metrics
>
> Our evaluation metrics, including cosine similarity of chord embeddings and OA of features, have also been applied by other papers in the literature. [1, 2, 3, 4] We appreciate the reviewer’s suggestion of adding subjective evaluations. While having a comprehensive comparison study of subjective evaluations is challenging, we plan to explore ways of adding sensible subjective evaluations or other non-quantitive alternative analysis for music.
>
> **References**
>
> [1] Ziyu Wang, Lejun Min, and Gus Xia. Whole-song hierarchical generation of symbolic music using cascaded diffusion models. The Twelfth International Conference on Learning Representations, 2024.
>
> [2] Ang Lv, Xu Tan, Peiling Lu, Wei Ye, Shikun Zhang, Jiang Bian, and Rui Yan. Getmusic: Generating any music tracks with a unified representation and diffusion framework. arXiv preprint arXiv:2305.10841, 2023.
>
> [3] Dimitri von Rutte, Luca Biggio, Yannic Kilcher, and Thomas Hofmann. Figaro: Controllable music generation using learned and expert features. In The Eleventh International Conference on Learning Representations, 2023.
>
> [4] Kristy Choi, Curtis Hawthorne, Ian Simon, Monica Dinculescu, and Jesse Engel. Encoding Musical Style with Transformer Autoencoders. In Proceedings of the 37th International Conference on Machine Learning, 2020.

---

> > ### Comment · Reviewer_aY42 · 2024-11-26
> >
> > I appreciate the authors’ prompt response and willingness to make changes and revisions to the manuscript. The clarifications and updates, particularly around task definition, theoretical analysis, and evaluation metrics, are appreciated. However, I believe some important aspects remain unresolved:
> >
> > 1. **Metrics and the “Wrong” Notes Debate**:
> >    Your response about providing control over the inclusion of out-of-key notes is compelling. While user-defined sampling guidance rules are helpful, it would be more interesting to explore whether the model could implicitly learn to generate contextually appropriate “wrong” notes. This could open valuable directions for future work, especially in genres where such surprises are intrinsic to the musical style. Otherwise, removing “wrong notes” seems like a straightforward task that could be achieved with simpler rule-based filtering added to a generative model (e.g., ensuring no “wrong notes” are generated and retrying if they are).
> >
> > 2. **Triple Meter Limitation**:
> >    Your explanation about the scarcity of 3/4 meter pieces in the dataset is reasonable. Thank you for clarifying!
> >
> > 3. **Theoretical Analysis**:
> >    Simplifying the theoretical section is a positive step, and the intuitive explanation is helpful. However, the question of whether this analysis is directly useful for real musical data remains unanswered. Still, It’s unclear why this remains a central focus when the proposed solution simply removes these notes.
> >
> > 4. **Classifier-Free Guidance (CFG)**:
> >    Thank you for clarifying the use of classifier-free guidance (CFG). From what I understand, CFG is essentially used as a switch between unconditional and conditional generations, which is, i guess, a valid way to use it. However, CFG is also widely applied in balancing diversity and quality in tasks like image generation. Did you experiment with CFG weights greater than 1? It would be interesting to see how the model performs at higher CFG weights and to compare the resulting audio with lower weights to evaluate the trade-off between diversity and stability.
> >
> > 5. **Metrics Evaluation**:
> >    I appreciate the references to prior works using the same metrics. However, I have a serious concern about the numbers in **Table 1** compared to the previous revision. As I recall (and as other reviewers also noted), your method did not show better results in OA(duration) and OA(note density) in the earlier version. How is it that these numbers are now better in the revised version? I did not see this explained anywhere. Could you clarify whether this change reflects a correction, additional experiments, or something else? Transparent justification for such changes is critical.
> >
> > 6. **Dataset and Task Definition**:
> >    The updated title and clarified scope of your method are improvements. However, ambiguity persists in how accompaniment generation and broader symbolic music generation tasks are defined and compared. For both tasks, it seems the work requires separate threads of logic, comparison to baselines, and so on. If you are claiming that your method works equally well for both accompaniment and melody generation tasks, I would expect comparisons with baselines in each domain, or alternatively, with a unified baseline that solves all these tasks. Right now, this distinction remains unclear in the paper.
> >
> > Again, I thank the authors for their work and their effort in promptly addressing feedback. However, considering the points outlined above, I do not plan to change my decision at this time.

---

> ### Author Response · Authors · 2024-11-28
>
> Thank you for the time and comments, and we appreciate your encouraging message on our response about providing user-defined control and intuitive explanation of the theory.
>
> We have added an initial set of **subjective evaluations** in Section 5.1.4 and Appendix F, and some additional **numerical experiments** in 5.1.5 and Appendix H of the current manuscript. We uploaded more examples on the demo page. Regarding your feedback about remaining unresolved aspects, we hope to add the following comments:
>
> **Metrics and the “Wrong” Notes Debate**
>
> Thank you for the insightful discussion. We agree that developing a model that enjoys the ability to implicitly learn contextually appropriate "wrong notes" would be immensely valuable. We believe that this would be an important line of future work, and we really look forward to the progress that can be made in this area. We think that this line of exploration would greatly complement, rather than conflicts with, the objectives of our current work. In addition, we think it is also worthwhile to develop methods that enable the model to function as a well-controlled collaborative tool to aid human composers. We believe that both directions are compelling and hold great potential for advancing the field.
>
> We have revised the descriptions and logic of our work to emphasize "control" rather than "correctness." Out-of-key notes are not inherently "wrong"; instead, the issue of degraded quality in generating out-of-key notes serve as an illustrative example of the challenges that current models face in accurately capturing distributional information in symbolic music data. Furthermore, the external sampling control is not limited to correcting "wrong notes", but is designed to provide a more efficient and reliable framework for aligning the model's output with the user's intent.
>
>
> We further hope to make the following clarifications:
>
> 1. Our method is intrinsically different from “detecting wrong notes and regeneration” in terms of sampling efficiency. There is no guarantee that regeneration is capable of eliminating all undesirable notes within a controllable amount of time. The improvement of efficiency has a vital benefit for real-time situations such as improvisation.
>
> 2. Different from straightforwardly removing “wrong notes”, our method is not targeted on “deleting incorrect notes in the final sample”. As described in the paper, our method fundamentally guides the diffusion sampling process by adjustments in middle and intermediate steps. If we view the diffusion process as a Stochastic Differential Equation (SDE), our method adjusts the drift term while preserving the stochasticity of the volatility term. This ensures that we balance both accuracy and the creativity of the diffusion model.
>
> Additionally, our method provides an effective controllable generation method with user-specified conditions of music. The method may be used to ensure certain inappropriate notes do not appear, and may also be used to maintain a pre-specified music type preferences. The efficient control by our method also enables the model to generate music of different styles and modes, even if such modes are not contained in its training set. We present such samples in Section 1 of our demo page.

---

> > ### Author Response · Authors · 2024-11-28
> >
> > **Theoretical Analysis**
> >
> > 1. Whether this analysis is directly useful for real music data:
> >
> > To clarify and correct possible misunderstanding, we like to note that the theory does not solely focus on the intrinsic patterns of real music data (e.g., high dimensionality). More importantly, it emphasizes the characteristics of existing generative models. Specifically, the term $\sup_{p \in P_\delta}$​ is taken over the potential **search space** of the generative model, where each specific parameterization of the neural network corresponds to an instance p in the distribution class $P_\delta$. Although music data is discrete, many state-of-art works [1,2] suggest using continuous diffusion models as estimators for the data distribution, thus resulting in a **search space of continuous probability distributions** for the model. Therefore, our theoretical analysis can directly apply to these situations.
> >
> > One remaining discussion may be on whether $P_\delta$ is exactly the same as the distribution class that we would encounter in real life. While real life distribution classes are more complicated and difficult to analyze, $P_\delta$ essentially captures their characteristics, and is therefore comparable to them. This type of simplification while maintaining core characteristics appears to be standard in works that provide theoretical insights.
> >
> >
> >
> > 2. Why this remains a central focus when the proposed solution simply removes these notes:
> >
> > We would like to clarify that the discussion of challenges in modeling error actually provides direct motivation for enhancing control over symbolic music generation.
> > The theoretical insights presented in Proposition 1 demonstrate that the occurrence of “unwanted notes” is often unavoidable, and the decay rate of this error probability with respect to dataset size is slow ($O(n^{−1/KL})$). Thus, relying on the model itself for precision is challenging for existing models, given the inherent scarcity of high-quality data and the slow decay rate of errors.
> > Thus, the implication of proposition 1, together with the fact that symbolic music generation requires an exceptional level of precision, motivates and highlights the contribution of our control mechanism, which offers an effective solution in achieving harmonic consistency. We hope this clarification strengthens the alignment between the theoretical discussion and the motivation for our methodology.
> >
> >
> > **Classifier-Free Guidance (CFG)**
> >
> > Thank you for your suggestions about evaluating how the model performs at higher CFG weights and to compare the resulting audio with lower weights to evaluate the trade-off between diversity and stability.
> >
> > We added a few experiments and analysis about the effect of different guidance weights in Appendix H of our new manuscript. We run experiments with guidance weights ranging from 0.5 to 10, with the results summarized in Table 5. Our findings imply that the model achieves higher sample quality for lower guidance weights, while moderately increasing the guidance weight makes the samples more adherent to the provided rhythmic conditions. However, excessively high guidance weight significantly reduces sample quality, and unnatural outputs undermine adherence to the rhythmic conditions.
> >
> > These findings are coherent with the results about the trade-off between diversity and stability in existing literature. Due to page limits and that this is not the main purpose of our paper, we decided to put it into the appendix of our manuscript.
> >
> > **Metrics Evaluation**
> >
> > We are sorry about not updating this information in our previous response (in our first-round response, we only updated to the reviewer that raised the question about “our method did not show better results in OA(duration) and OA(note density)”). During the rebuttal session, we conducted additional experiments that requires retraining our models, and observed a change in the OA values. The improvement is most likely due to an increase in training epochs-from 10 in the previous version to 20 in the current experiments. The new number of epochs was updated in our manuscript.

---

> > > ### Author Response · Authors · 2024-11-28
> > >
> > > **Dataset and Task Definition**
> > >
> > > Thank you for your suggestions on melody generation tasks. We have updated some experiments in Appendix E.3 about symbolic music generation tasks (where both melody and accompaniment are generated) conditioning solely on chord progression. The results indicate that our FTG method still outperforms baselines on this task. Moreover, to better illustrate our method’s effectiveness in symbolic music generation, we uploaded some generated examples in Section 1 of our demo page. For all those examples, we only provided the model with chord and rhythm conditions. Also, as mentioned before, we controlled the sampling process on non-standard scales (including Chinese pentatonic scale and Dorian scale), and the generated samples turned out to fit well with the corresponding styles, even if they do not exist in the training set. This further highlights the power of our FTG method in symbolic music generation.
> > >
> > > However, compared with the accompaniment generation task, we remark that the chord-conditioned symbolic music generation experiment does not have enough effective basis for comparison, and that is the reason why we did not conduct this in our first version. Some reasons are follows:
> > >
> > > For the accompaniment generation task, the comparison with ground truth on features (such as pitch, duration, and note density) make sense, because the leading melody inherently contains many constraints on the rhythm and pitch range of the accompaniment, ensuring coherence with the melody. Thus, similarity with ground truth on those metrics serves as an indicator of how well the generated samples adhere to the melody.
> > >
> > > However, in symbolic music generation conditioned only on a chord sequence, while chord progression similarity remains comparable (as the chord sequence is provided), evaluating features against ground truth is less informative. This is because multiple different pitch range and rhythm could appropriately align with a given chord progression, making deviations from the ground truth in these features less indicative of sample quality. Therefore, chord similarity emerges as the sole applicable metric in this context.
> > >
> > > Additionally, WholeSongGen's architecture does not support music generation conditioned solely on chord progressions, as it utilizes a shared piano-roll for both chord and melody, rendering it unsuitable for comparison. Conversely, GETMusic facilitates the generation of both melody and piano accompaniment based on chord conditions, allowing for a viable comparison. Consequently, we only present results focusing on chord similarity between our model and GETMusic. The results indicate that our controlled FTG method achieves better performance than both GETMusic and FTG without sampling control.
> > >
> > >
> > > **Conclusion**
> > >
> > > We hope our response addresses your concerns. Once again, we sincerely appreciate your timely and valuable feedback, which has inspired us to reconsider in depth the potential applications and implications of our method. Your suggestions have greatly helped us refine our presentation, including the design of numerical experiments, the appropriateness of word choice, and the overall arrangement and logic of our manuscript. We are really happy to see our work taking a more polished form.
> > >
> > > **References**
> > >
> > > [1] Ziyu Wang, Lejun Min, and Gus Xia. Whole-song hierarchical generation of symbolic music using cascaded diffusion models. The Twelfth International Conference on Learning Representations, 2024.
> > >
> > > [2] Yujia Huang, Adishree Ghatare, Yuanzhe Liu, Ziniu Hu, Qinsheng Zhang, Chandramouli S Sastry, Siddharth Gururani, Sageev Oore, and Yisong Yue. Symbolic music generation with non-differentiable rule guided diffusion

---

### Official Review · Reviewer_jQkk · 2024-10-30

**Soundness:** 3
**Presentation:** 3
**Contribution:** 2
**Rating:** 5
**Confidence:** 3

**Summary:**

This paper addresses a common issue in generative music models: the tendency to produce out-of-key notes without adequate contextual grounding, which derives noticeable dissonance. It presents a theoretical explanation for why such errors may persist even in well-trained models. To solve this problem, this paper proposes Fine-Grained Texture Guidance (FTG), a sampling strategy used in the reverse process of a diffusion model. FTG first identifies out-of-key notes (i.e., pixels on a piano roll image) based on the local key signature. It further reduces their likelihood at each sampling step, thus preventing high-probability outputs at those positions. Experimental results on the accompaniment generation task (given melody and chords) demonstrate that FTG effectively improves harmonicity, surpassing existing diffusion-based approaches.

**Strengths:**

This paper provides a theoretical analysis on the inevitability of deriving out-of-key notes by music generative models. This provides a good motivation for the technical part of this paper.

**Weaknesses:**

* While FTG targets eliminating out-of-key notes, there seems to be a trade-off between harmonicity and creativity. The demo accompaniments mostly consist of sparse broken chords, with a lower degree of polyphony than ground-truth pieces in POP909. In other words, the piano textures are less well-formed. The reviewer thus wonders if the proposed idea (eliminating out-of-key notes) makes sufficient musical sense. The evaluation results in Table 1 seem to support the reviewer's view. Specifically, FTG earns higher Chord Similarity and OA(pitch), both of which are related to harmony; however, it falls short in OA(duration) and OA(note density), which are related to rhythm.

* Experiments in this paper primarily involve 4-bar music clips with a local key signature given. In practice, we could see longer pieces, and it is non-trivial to detect key variations within a piece. Given that key variations such as modal interchange are common in pop music, this could be a limitation which barriers the practical application of the FTG method.

* Experiments in this paper are insufficient in the following aspects:

   1. Method-wise, there lacks an ablation study on the performance (of the same diffusion model) with and without FTG sampling. This could be critical in helping readers understand the impact of "eliminating out-of-key notes." $ \color{red} \mathrm{addressed} $

   2. Task-wise, there lacks other (non-diffusion) baselines for the accompaniment arrangement task [1,2].

   3. Besides the objective evaluation metrics, subjective experiments could be necessary to evaluate music quality and creativity. $ \color{red} \mathrm{addressed} $

[1] J. Zhao, et al. Accomontage: Accompaniment arrangement via phrase selection and style transfer, in ISMIR 2021.

[2] S.-L. Wu and Y.-H. Yang. Compose & Embellish: Well-structured piano performance generation via a two-stage approach, in ICASSP 2023.

**Questions:**

*  Based on the reviewer's understanding, FTG is an inference-time strategy, which can be applied in an ad hoc manner to a pre-trained diffusion model. Can it be directly applied to WholeSongGen? Similarly, it could be helpful to conduct an ablation study to test the diffusion model developed in this paper with and without FTG. $ \color{red} \mathrm{addressed} $

*  Lines 505-508 describe the calculation of chord accuracy, which is based on the similarity of latent chord features. How about directly comparing the chord sequences at the observational level? $ \color{red} \mathrm{addressed} $

---

> ### Author Response · Authors · 2024-11-26
> **Response to Reviewer jQkk**
>
> We thank the reviewer for the valuable suggestions. We have modified our maniscript and have the following comments:
>
> 1. The trade-off between harmonicity and creativity
>
> While the out-of-key notes are never necessarily “wrong”, it is observed that generative models often fail to create an accommodating context for such “creativity”. We have uploaded examples to part 1 of the demo page (https://huggingface.co/spaces/interactive-symbolic-music/InteractiveSymbolicMusicDemo) to demonstrate the phenomenon. Sometimes the generated results without control can be completely chaotic (e.g., example 1 and 3 in section 2 of our demo page).
>
> Further, we have retrained our model (in the previous version we only run 10 training epochs, now we increase the number of epochs to 20), and now the OA score for duration and note density have become higher than WholeSongGen. Third, the broken chords are not a result of removing out-of-key notes, but also appear in the original training data.
>
> Moreover, we additionally remark that our method is not targeted on “deleting incorrect notes in the final sample”. Instead, our method provides an effective controllable generation method with any user-specified conditions of music. The method may be used to ensure certain inappropriate notes do not appear, and may also be used to maintain a pre-specified music type preferences. As described in our paper, our method fundamentally guides the diffusion sampling process by adjustments in middle steps. If we view the diffusion process as a Stochastic Differential Equation (SDE), our method adjusts the drift term while preserving the stochasticity of the volatility term. This ensures that we balance both accuracy and the creativity of the diffusion model.
>
> Effectively, during the diffusion steps, our approach does not just remove incorrect notes but also actively guides the model to generate more harmonically appropriate notes. This guidance process helps the model converge to musically valid outputs while still leveraging the generative freedom of the diffusion model.
>
> To further illustrate this, in Figure 6 of our manuscript, we provide two concrete examples demonstrating that our sampling control not only removes incorrect notes but also encourages the generation of notes that are more harmonic and contextually appropriate. Additionally, Section 2 of our demo page (https://huggingface.co/spaces/interactive-symbolic-music/InteractiveSymbolicMusicDemo) provides samples showing how even completely chaotic results (e.g., Example 1) can be effectively guided back into musically valid outcomes (rather than removing everything) through our method.
>
> 2. “In practice, we could see longer pieces, and it is non-trivial to detect key variations within a piece. Given that key variations such as modal interchange are common in pop music, this could be a limitation which barriers the practical application of the FTG method.”
>
> Unfortunately this is due to a misunderstanding of the reviewer. First, even in the 4-bar phrases we actually apply different keys, which are associated to chords. The “key signature” here does not refer to the key signature of a whole piece, but rather the key associated with the temporary tonic, as we have explained in section 3.1.
>
> Considering longer generations, we can actually apply the inpainting methods of e.g., WholeSongGen[1] to create longer music out of 4-bar sections, but this is not the focus of this paper. Our main focus is to propose the precise controlling method for diffusion models, which can in fact be applied to other diffusion-based symbolic music generation methods.
>
> **References**
>
> [1]Ziyu Wang, Lejun Min, and Gus Xia. Whole-song hierarchical generation of symbolic music using cascaded diffusion models.
>
> [2]Yujia Huang, Adishree Ghatare, Yuanzhe Liu, Ziniu Hu, Qinsheng Zhang, Chandramouli S Sastry, Siddharth Gururani, Sageev Oore, and Yisong Yue. Symbolic music generation with non-differentiable rule guided diffusion

---

> ### Author Response · Authors · 2024-11-26
>
> 3. About ablation study
>
> We thank the reviewer’s suggestion of adding ablation studies. We have added an ablation study section (Section 5.1.5) to our manuscript. We run two additional experiments on the same accompaniment generation task to analyze the impact of the fine-grained conditioning during training and the fine-grained control in sampling. The first experiment involves the same model trained with fine-grained conditioning but without control during sampling, while the second is an unconditional model without any conditioning or control in both the training and sampling process. We evaluate the frequency of out-of-key notes and assess the overall performance using the same quantitative metrics as before. We find that conditioning in training and control in sampling both contribute to reducing out-of-key notes and improving overall performance.
>
> Moreover, in Figure 6 of our manuscript, we provide two examples showing that our sampling control can not only eliminate incorrect notes, but also guide the model to replace them with more harmonic ones.
>
> 4. About comparison with other (non-diffusion) models and subjective evaluation
>
> For comparing with other (non-diffusion) models, we did not conduct such experiments because our focus in this work is to provide a control methodology based on diffusion-based symbolic generation models, not to prove that diffusion models perform better than other models, which has been discussed in many works on diffusion-based models, such as [1] and [2]
>
> As for subjective evaluation, we appreciate the reviewer’s suggestion of adding subjective evaluations. While having a comprehensive comparison study of subjective evaluations is challenging, we plan to explore ways of adding sensible subjective evaluations or other non-quantitive alternative analysis for music.
>
> 5. Can FTG be applied to WholeSongGen?
>
> We first clarify that FTG not only contains ad hoc sampling control, but also contains pixel-wise conditioning for chord and rhythms in the training process. Therefore, it cannot be directly applied to WholeSongGen, but can be integrated into the framework of WholeSongGen with modifications and combinations. Generally, our idea of fine-grained control in the sampling process can be applied (with modifications) to other diffusion-based models for symbolic music generation.
>
> 6. Metric of chord accuracy in latent space
>
> Designing a direct comparison metric for chord sequences—such as determining whether "D" or "E minor" is closer to "C" on the observational chord-wise basis—is inherently complex. Therefore, we prefer the cosine similarity of the latent chord feature because the distances of latent vectors are comparable. In fact, our metric of comparing the latent chord feature has already been applied in the literature, such as [1].
>
> **References**
>
> [1] Ziyu Wang, Lejun Min, and Gus Xia. Whole-song hierarchical generation of symbolic music using cascaded diffusion models.
>
> [2] Yujia Huang, Adishree Ghatare, Yuanzhe Liu, Ziniu Hu, Qinsheng Zhang, Chandramouli S Sastry, Siddharth Gururani, Sageev Oore, and Yisong Yue. Symbolic music generation with non-differentiable rule guided diffusion

---

> > ### Comment · Reviewer_jQkk · 2024-11-29
> >
> > Thank you for addressing my review.
> >
> > I appreciate the authors' efforts in responding to my concerns regarding the ablation study and subjective experiment. These issues have been addressed, and I have accordingly increased my score from 3 to 5.
> >
> > However, I would like to revisit my question about key signatures. I understand that local key signatures, rather than a global key, are employed in the proposed method. Could you clarify how these local key signatures are detected? My concern is that in longer pieces with key variations, accurately identifying local key signatures and their boundaries could be challenging, thus potentially limiting the applicability of the proposed method in practical scenarios.
> >
> > Regarding the listening demos, I still perceive a noticeable tendency towards a lower degree of polyphony, rhythmic creativity, and overall naturalness compared to the ground-truth POP909 dataset or the outputs from Whole-Song-Gen and GETMusic.
> >
> > Lastly, while I understand the rationale for limiting comparisons to diffusion-based models in order to narrow the scope of the paper, this choice may also restrict the potential impact of the work. A broader comparison could strengthen the significance, reliability, and reach of your contributions.

---

> ### Author Response · Authors · 2024-11-29
>
> Thank you very much for encouraging and recognizing our effort.
>
> Regarding the question about local key signatures, we would like to clarify that they are not "detected" but rather "designed" by the user, and provided to the model in the generation process.
>
> To explain, two chromatic elements are included in the generation processes of our model: chord and local key regularization.
>
> - **Chord**: During training, chords are inferred (either through standard chord recognition algorithms or human labeling) from the music, and provided as conditions. In the generation process, the user specifies a desired chord sequence, which the model uses as a condition to generate music based on it.
>
> - **Local Key Regularization**: Unlike chords, the local key is not included in the training process. Instead, it serves as an additional user-defined input during generation, designed to better regularize the model's output. By specifying a local key, the user restricts the pitch classes that the model can use in its output, effectively shaping the generated music to align with a desired tonal structure.
>
> For example, if the user wishes to generate a measure aligning with the chord $C$, they might choose the $C$ major scale as the "key regularization" for this measure. In this case, the user restricts the model from generating pitch classes $C\sharp$, $D\sharp$, $F\sharp$, $A\flat$, and $B\flat$ within that measure. Alternatively, the user could use the Chinese pentatonic scale as the "key regularization" for the measure, in which case the model would only output the pitch classes $C$, $D$, $E$, $G$, and $A$. This flexibility allows users to customize the tonal structure of the generated music. This feature has not yet been integrated into the "user interface" of our demo page—not because it is technically challenging, but simply due to time constraints.
>
> While the term "local key" may initially seem misleading, it was introduced in earlier discussions about the challenges in precision in symbolic music generation. Specifically, when models generate notes outside the "local key", the resulting sample quality often degrades. Later, in the context of applying the FTG method, the design of every local "key regularization" is left entirely up to the user. A common choice is to use a scale that contains the pitch classes of the chord condition.
>
> We hope that this clarifies the role of "local key" or "local key regularization" in our work. Again, thank you for the comments and discussion.

---

### Official Review · Reviewer_emmS · 2024-11-03

**Soundness:** 1
**Presentation:** 2
**Contribution:** 2
**Rating:** 3
**Confidence:** 2

**Summary:**

This paper proposes two methods aimed at encouraging piano-roll diffusion models to produce conditioned generations that adhere to given key signatures, chords, or rhythmic structures. First, the authors train a diffusion model conditioned on rhythmic and harmonic masks, and use classifier-free guidance to control the guidance level. Second, they introduce an inference-time technique that adjusts noise predictions according to given piano-roll masks.

The authors evaluate their approach at accompaniment generation by extracting chord progressions from generated samples and comparing the cosine similarity of latent space embeddings with those derived from the ground truth. They also compare statistical features of the generations, such as note-pitch distributions with the ground truth.

**Strengths:**

Since I am not an expert in diffusion models, I cannot confidently assess the correctness of Proposition 1 or the novelty of the inference technique proposed in Section 4.2. However, I have identified the following strengths:

1) Novelty in symbolic music generation: The authors introduce an innovative approach for guiding diffusion models by using classifier-free guidance to condition generation on piano-roll masks. To my knowledge, this guidance method is new within the context of symbolic piano-roll generation.

2) Potential impact: Although this work focuses on chord/key adherence, the introduced techniques could potentially be adapted for more nuanced conditioning in future research.

**Weaknesses:**

Since I am not an expert in diffusion models, I will refrain from commenting on the novelty and correctness of the proposed approaches and will limit my comments to the evaluation section, which is fairly limited.

1) The objective evaluations and metrics are quite limited, making it difficult to draw strong conclusions from this paper. Although the authors have clearly put significant effort into the demo page, it cannot be relied upon from a scientific perspective. Relevant experiments and ablations are missing, such as counting the number of generated out-of-key notes across different techniques.

2) No ablation studies are conducted. Given that the authors introduce two separate techniques, it would be reasonable to compare them in Section 5.2. It would also be useful to compare both techniques with a baseline unconditioned diffusion model.

3) In Section 5.2.5, Table 1, the introduced FTG approach is compared against GETMusic and WholeSongGen; however, it is unclear how these comparisons are relevant. To my knowledge, GETMusic cannot be conditioned on control signals for key or chords. The main takeaway from Section 5 is, therefore, that by conditioning a model on the same chord progressions as the ground truth, the chord and pitch similarity (compared to the ground truth) improves.

4) This work is mathematically dense, and the appendix is extensive. While I am not an expert in diffusion models, I wonder if this work could be reorganized to place greater emphasis on numerical evidence and less on mathematical motivation.

5) Restricted scope: The work could be improved by testing the proposed inference technique in more varied domains.

**Questions:**

1) I am particularly interested in how the proposed guidance approaches compare to each other and to other established methods. Did you run these tests?

2) Did you consider or compare to a more straightforward approach: guiding the diffusion model by directly masking the undesirable parts of the piano roll? There appears to be significant literature on this topic for image diffusion models, e.g., [1].

[1] Lugmayr, A., Danelljan, M., Romero, A., Yu, F., Timofte, R., & Van Gool, L. (2022). Repaint: Inpainting using denoising diffusion probabilistic models. In Proceedings of the IEEE/CVF Conference on Computer Vision and Pattern Recognition (pp. 11461–11471).

---

> ### Author Response · Authors · 2024-11-26
> **Response to Reviewer emmS**
>
> We thank the reviewer for the valuable suggestions. We have modified our manuscript and have the following comments:
>
> 1. Objective evaluations and experiment details:
>
> Experiments and ablation study: We thank the reviewer’s suggestion of adding ablation studies. We have added an ablation study section (Section 5.2.5) to our manuscript. We run two additional experiments on the same accompaniment generation task to analyze the impact of the fine-grained conditioning during training and the fine-grained control in sampling. The first experiment involves the same model trained with fine-grained conditioning but without control during sampling, while the second is an unconditional model without any conditioning or control in both the training and sampling process. We evaluate the frequency of out-of-key notes and assess the overall performance using the same quantitative metrics as before. The results indicate that conditioning in training and control in sampling both contribute to reducing out-of-key notes and improving overall performance.
>
> Moreover, in Section 5.1 of our manuscript, we provide two examples showing that our sampling control can not only eliminate incorrect notes, but also guide the model to replace them with more harmonic ones.
>
> 2. About the comparison with GETMusic
>
> As for the comparison with GETMusic, GETMusic in fact has an input channel for conditional chord sequences, see “(6) chord” on github repo https://github.com/microsoft/muzic/tree/main/getmusic. It also has a channel for melody. Therefore, in our evaluation section, all models (ours, WholeSongGen, GETMusic) are utilizing chord and melody conditions to generate accompaniments, which ensures that they are comparable.
>
> 3. About dense mathematical notations
>
> We are sorry about the dense mathematical notations. We have revised our manuscript to be more straightforward, shortened the mathematical part, and added more empirical studies.
>
> 4. About comparison with each other and other methods
>
> As mentioned in comment 1, we have added an ablation study section (Section 5.2.5) to our manuscript. As for the comparison with other established methods, we have already provided the comparison with WholeSongGen[1] and GETMusic[2] in Section 5.2.4 on the accompaniment generation task.
>
> 5. About "masking and repainting"
>
> We did not consider repaint because it requires multiple times of resampling, which does not achieve the desired sampling efficiency as in this work. Further, there is no guarantee that repaint is capable of eliminating all undesirable notes within a controllable amount of time. In contrast, our sampling guidance guarantees that undesirable notes could be avoided in one sampling process, and the controls in middle sampling steps (rather than only in the last step) could retain the flexibility and creativity of diffusion models.
>
> **References**
>
> [1] Ziyu Wang, Lejun Min, and Gus Xia. Whole-song hierarchical generation of symbolic music using cascaded diffusion models. The Twelfth International Conference on Learning Representations,2024.
>
> [2] Ang Lv, Xu Tan, Peiling Lu, Wei Ye, Shikun Zhang, Jiang Bian, and Rui Yan. Getmusic: Generating any music tracks with a unified representation and diffusion framework. arXiv preprint arXiv:2305.10841, 2023.

---

> ### Comment · Reviewer_XuiB · 2024-11-26
>
> Thanks for your rebuttal.
>
> I truly appreciate your revision to the manuscript and it does improve the paper quality. After discussion, I can feel that the main issue of this paper is its readability, so that we need a round of rebuttal to clarify some necessary details.
>
> I admit that the theoretical part of this paper provides useful insights, which is a quite interesting section and is also coherent with previous intuitions. Then the last main issue would be sufficient **subjective experiments**. I totally understand that there is no time for experiment during rebuttal period, but it is the golden test for this kind of task.
>
> Also, in my review I mentioned that it would be helpful if you can compare the performance with methods that also use **control methods**.
>
> I am happy to raise the score to 8, assuming you will complete these experiments.

---

> > ### Author Response · Authors · 2024-11-26
> >
> > Thanks for the discussion and for giving us the opportunity to clarify the necessary details.
> >
> > We appreciate the reviewer’s encouragement and recognition on the theoretical part and the associated insights/coherence.
> >
> > We appreciate the suggestion on comparing with methods that also use control methods. We concur with this aspect of comparison. We would like to bring up that the experiments in Section 5.2.4 of the current version contain some flavor of this aspect. Specifically, we compared our method with WholeSongGen, which has control on music phrases, lead melody, and chord progression information, and GETMusic, which generate piano tracks with chord and/or melody conditions as external control. Therefore, in our Section Section 5.2.4, all models (ours, WholeSongGen, GETMusic) are already controlled, which ensures that they are comparable.
> >
> > Indeed, we feel that adding sensible **subjective evaluations** is going to make the work’s overall experiments more comprehensive. We have been working on a set of blind tests with subjective evaluators, with fair design of questions. We will add our results as soon as possible.
> >
> > We would love to thank the reviewer again for the valuable comments and suggestions.

---

> ### Comment · Reviewer_emmS · 2024-11-27
>
> Thank you for taking the time to address my concerns.
>
> The revised manuscript is much improved, and the additional ablation experiments are welcome.
>
> Concurring with other reviewers, I still have doubts about the significance of this work and its evaluations. To properly evaluate this work, it's important to add additional experiments, particularly subjective comparisons with other approaches for conditioning diffusion models. As it stands, it's quite hard to evaluate whether this approach works better than rule-based filtering, or even just manually removing masked notes as a post-processing step, within the presented evaluations. I can't rely on the provided samples to evaluate the contribution of this work alone, as I don't know to what degree the samples were cherry-picked. Additionally, the output from the unconditioned model sounds much less musical than other approaches to accompaniment generation. This raises questions about whether training on the POP909 (which is relatively small) is really appropriate if the goal is to faithfully measure the impact of guidance on musical generation quality, which appears to be what the authors are trying to demonstrate with the demo page.
>
> As I am unable to properly evaluate the paper's contributions, and am unsure if the proposed methods have any applications beyond efficiently conditioning data-scarce piano-roll diffusion models for accompaniment generation, I recommend rejecting the paper in its current form.
>
> I must stress to the AC that as I am not an expert in diffusion models or piano-roll generation using diffusion models, I've selected a confidence of 2.

---

> ### Author Response · Authors · 2024-11-28
>
> We appreciate the reviewer’s recognition of our previous round of responses, the improvement of manuscript and the additional experiments.
>
> In addition to the previous round of responses and the revision made then, we have made the following changes to the latest version of our manuscript:
>
> 1. In Section 5.1.4, we added an initial group of **subjective evaluations**. We presented average ratings and confidence intervals for six different aspects (details described in Appendix F). The results suggested that our method outperforms the baselines in 6 different aspects.
> 2. In section 5.1.5, we directly compared our method with an additional baseline method about “just manually removing masked notes as a post-processing step”. Our sampling control method outperformed this baseline.
> In Section 1 of our demo page, we have added several examples showing the ability of our model in generate music of different styles and genres, even if such genres are not contained in its training set.
>
> We have the following comments the recent response from reviewer emmS:
>
> *“As it stands, it's quite hard to evaluate whether this approach works better than rule-based filtering, or even just manually removing masked notes as a post-processing step, within the presented evaluations.”*
>
> **Response**: We further hope to make the following clarifications:
>
> 1. First, we respectfully ask the reviewer not to neglect, in additional to the sampling part of contribution, the other parts of our approach, for example the fine-grained training condition.
>
> 2. As for the sampling control, in our ablation study in Section 5.1.5, we added a comparison with the method of “post-sample rule-based editing”, where we numerically demonstrate that the method of inserting control in the sampling process is essentially different from rule-based post-sample editing.
> We additionally remark that our method is not targeted on “deleting incorrect notes in the final sample”. Instead, our method provides an effective controllable generation method with any user-specified conditions of music. The method may be used to ensure certain inappropriate notes do not appear, and may also be used to maintain a pre-specified music type preferences. As described in our paper, our method fundamentally guides the diffusion sampling process by adjustments in middle and intermediate steps. If we view the diffusion process as a Stochastic Differential Equation (SDE), our method adjusts the drift term while preserving the stochasticity of the volatility term. This ensures that we balance both accuracy and the creativity of the diffusion model, leveraging the structured gradual denoising process of diffusion models to preserve distributional properties of the original learned distribution. In contrast, rule-based editing, different from our approach, would employ a brute-force editing approach that disrupts the generated samples, disrupting melodic lines and rhythmic patterns. Both the theoretical guarantee provided by proposition 2 and the numerical results further validate this analysis.
>
>
> 3. As stated in our manuscript, our method is intrinsically different from “detecting wrong notes and regeneration” in terms of sampling efficiency. Also, there is no guarantee that regeneration is capable of eliminating all undesirable notes within a controllable amount of time. The improvement of efficiency has a vital benefit for situations such as improvisation.
>
>
> 4. Further, far different from straightforwardly removing “wrong notes” or "detecting wrong notes and regeneration", our method also enables the model to **generate music of different styles and genres**, even if such genres are not contained in its training set. We present such samples in Section 1 of our current demo page. Those examples demonstrate the power and flexibility of method.

---

> ### Author Response · Authors · 2024-11-28
>
> *“Additionally, the output from the unconditioned model sounds much less musical than other approaches to accompaniment generation. This raises questions about whether training on the POP909 (which is relatively small) is really appropriate if the goal is to faithfully measure the impact of guidance on musical generation quality, which appears to be what the authors are trying to demonstrate with the demo page.”*
>
> **Response**: As a clarification, what we are trying to demonstrate on the demo page is for the purpose of responding to the question “if eliminating 'wrong' notes entirely could risk making the output less interesting”. The samples on the demo page are not the general case, but specifically cases containing out-of-key notes, to support our statement: that there can be cases where model outputs are very unreliable, which provides motivation for our method providing strong control to improve the reliability of models.
>
> We emphasize that the outputs from the uncontrolled model serves as an ablation study. The examples on the demo page shows that, even if a sample from the uncontrolled model is that unmusical, our sampling control can still guide it to a much more musical one. Such an improvement indeed emphasizes the power of our sampling control, and shows that the sampling control is an irreplaceable part of our method.
>
> As for the question of using POP909, we note that this dataset was the only training dataset for literature like [1] “Whole-song hierarchical generation of symbolic music using cascaded diffusion models (ICLR 2024)”, one of the baseline methods. This partly suggest the appropriateness of using this dataset. The other part is that, there are very few symbolic music datasets that have the most structured data and clear labels. This is again, rooted in the facts that high-quality symbolic music data are genuinely scarce compared to some other data modalities, which in turn, further supports our work’s motivation and contribution to improve music generation quality via external control other than increasing data.
>
>
> *“I can't rely on the provided samples to evaluate the contribution of this work alone, as I don't know to what degree the samples were cherry-picked.”*
>
> **Response**: This comment can be made to all music generation papers to question whether the samples are “cherry-picked”. We do not think it is fair to raise concerns to our work about “cherry-picking” without discrediting effectively all the generation work in this domain.
>
> In fact, to the best of our knowledge, we are **the only work in symbolic music generation** that provides a **user interface** on the demo page where **users can self-generate directly from the model**.

---

> ### Comment · Reviewer_emmS · 2024-11-29
>
> Hello,
>
> I really appreciate the author's eagerness and efforts to revise the manuscript and respond to feedback; however, in my opinion, this work is below ICLR standards. I have some final remarks which I hope will help improve the manuscript further.
>
> > In Section 5.1.4, we added an initial group of subjective evaluations. We presented average ratings and confidence intervals for six different aspects (details described in Appendix F). The results suggested that our method outperforms the baselines in 6 different aspects.
>
> The results displayed in Figure 3 appear to be quite marginal. Although no discussion of statistical significance is given, it seems very likely that these results are not statistically significant. This is below the scientific standards I would expect for a conference like ICLR.
>
> Moreover, I have serious concerns about the design of the study. If I understand correctly, the results show that the ground truth (i.e., real accompaniment) was judged equally to the generated accompaniment in the subjective evaluation? This makes me incredibly suspicious of the experimental design, given what I have heard on the demo page.
>
> The questions in the subjective evaluation are also not ideal. What does 'richness' mean in a musical sense? From what I can tell, the participants were not given sufficient information to answer these questions properly. The fact that every category follows the same distribution should raise serious questions about the experimental design in my opinion.
>
> > In section 5.1.5, we directly compared our method with an additional baseline method about "just manually removing masked notes as a post-processing step". Our sampling control method outperformed this baseline. In Section 1 of our demo page, we have added several examples showing the ability of our model in generating music of different styles and genres, even if such genres are not contained in its training set.
>
> If I understand correctly, the differences between the 'training and sampling control' and 'training control with edit' are *very* marginal. As I mentioned in my previous response, in order to properly evaluate the proposed methods in this work, there needs to be better ablation tests. Table 2 could also benefit from having a more descriptive caption.
>
> > This ensures that we balance both accuracy and the creativity of the diffusion model, leveraging the structured gradual denoising process of diffusion models to preserve distributional properties of the original learned distribution.
>
> I completely agree with this observation. The point I was raising is that the evaluations provided don't allow the reviewer to verify this claim.
>
> > We did not consider repaint because it requires multiple times of resampling, which does not achieve the desired sampling efficiency as in this work. Further, there is no guarantee that repaint is capable of eliminating all undesirable notes within a controllable amount of time. In contrast, our sampling guidance guarantees that undesirable notes could be avoided in one sampling process, and the controls in middle sampling steps (rather than only in the last step) could retain the flexibility and creativity of diffusion models.
>
> I find this response unconvincing. From what I can tell, there is a massive amount of literature concerning in-painting with diffusion models. The problem proposed in this work seems to very closely resemble an in-painting task, where a mask is specified over the out-of-key notes, i.e., signifying that these notes are not present and shouldn't be generated. I don't think sampling efficiency is a sufficient reason to omit this very natural comparison. From what I can see, modern in-painting techniques seem to retain the flexibility and creativity of diffusion models while forcing the diffusion process to leave some areas of the image/piano-roll unaffected.
>
> > First, we respectfully ask the reviewer not to neglect, in addition to the sampling part of contribution, the other parts of our approach, for example the fine-grained training condition.
>
> This is certainly a contribution in the context of piano-roll generations; however, I have doubts about the broader significance and novelty.

---

> ### Author Response · Authors · 2024-11-29
>
> In our previous responses to reviewer emmS, we included two parts. **First**, we responded with concrete additional experiment results that demonstrated positive comparative performance of our method. **Second**, we pointed out objectively that the reviewer’s comment *“I can't rely on the provided samples to evaluate the contribution of this work alone, as I don't know to what degree the samples were cherry-picked”* is **not fair** by the comment itself, and even more unfair given that we are the only work in symbolic music generation that provides a user interface on the demo page where users can self-generate directly from the model.
>
> Then the reviewer emmS chose to respond to our first part -- the positive additional experiments -- with very subjective and negative comments, using words like “quite marginal”, “incredibly suspicious”. The reviewer emmS chose to completely ignore the second part that provided evidence of the reviewer being unfair. Meanwhile, the reviewer reduced the rating to 3 in response to our positive additional experiments, which points out the fact that the reviewer is unfair.
>
> The above facts and evidence suggest that the reviewer emmS is unfair and biased. It is meaningless for a fair audience to engage with reviewer emmS, before reviewer emmS honestly admits to the biased and unfair behavior in front of the evidence.

---

> ### Comment · Reviewer_emmS · 2024-11-30
>
> Hello Authors.
>
> I'm very sorry if my comments were offensive or you found them to be unfair. That was not my intention. To add clarification based on the author's comments, my concerns about cherry-picking have been addressed. The inclusion of the user interface on the demo page is also certainly a good addition.
>
> I lowered my score because of my concerns about the evaluation. I would recommend to the AC to take my review into context, given the low confidence score.

---

### Official Review · Reviewer_XuiB · 2024-11-04

**Soundness:** 3
**Presentation:** 2
**Contribution:** 3
**Rating:** 8
**Confidence:** 4

**Summary:**

Certainly. Here is the revised review incorporating your additional point and overall refinement.

Summary

This paper aims to illustrate the challenges symbolic music generation models face in producing music with precise pitch and consistent rhythm, especially when trained on limited datasets. To address this, the authors explore applying multiple fine-grained controls in a diffusion-based model, specifically incorporating chord and rhythm controls during training and pitch constraints during sampling to reduce the generation of inharmonic notes. The experiments, centred on a 2-bar piano accompaniment generation task, indicate that the proposed approach improves alignment between the model output and ground truth for these features.

The methodology shows creativity, yet several concerns impact the overall cohesiveness of the paper.

Firstly, the introduction suggests that a limitation in symbolic music generation is the lack of high-quality training data and argues that current methods “fail to avoid” specific issues due to small datasets. However, this focus on data scarcity appears somewhat misplaced in a paper primarily about enhancing control within the generation process. Many large-scale methods now commonly address symbolic music generation challenges by scaling up data rather than solely refining model controls. Although data scarcity and control precision are not mutually exclusive issues, emphasising data limitations detracts from the paper’s central focus on control. Additionally, the authors’ use of the POP909 dataset for their experiments does constrain the model to some degree, but it does not fully justify the extensive discussion around data limitations, which might shift the reader’s focus away from the main topic of control mechanisms.

The paper provides theoretical explanations for why the models might fail to prevent certain errors, but this assumption feels somewhat strong. Specifically, the authors argue that continuous estimators, such as piano roll generation models, struggle with precision in small datasets. However, a diffusion model is a more complex, progressive structure. Although theoretically, a diffusion model could be viewed as a continuous estimator, prior research suggests that progressive generators have improved capacity for generation and representation over simpler continuous estimators like VAEs.

The approach centres on adding fine-grained controls during training, positioning the main contribution around the novelty and effectiveness of these controls. However, several existing studies also employ external controls—such as chord progressions, themes, and chromagrams—for music generation. The authors could clarify how their method compares or advances beyond these established control techniques. While external control indeed reduces the search space and improves model alignment with conditioning, more discussion on the specific advancements offered by this approach would strengthen the paper.

Additionally, the paper proposes applying continuous constraints during sampling to enhance pitch accuracy. However, this claim depends on several strong assumptions: that noise correction can consistently limit off-key notes, that fine-grained control will not disrupt local patterns, and that early stopping time ￼ can be practically achieved.

Given these assumptions, the study’s experiments are limited to 2-bar piano accompaniment generation with objective analysis only. This raises questions as to whether the theoretical discussions fully align with the experimental findings.

The most notable contribution of this paper may be the dynamic pitch correction during sampling; aside from this, other aspects, like chord control, might not necessitate such extensive discussion.

**Strengths:**

- The study’s main strength lies in its creative use of control techniques during both training and sampling phases.

- The approach to apply separate controls on chord, rhythm, and pitch demonstrates an innovative attempt at refining generated music towards a more structured and precise outcome, and the objective evaluation appears to validate improved consistency between generated music and ground truth features.

**Weaknesses:**

The authors place significant emphasis on data scarcity as a limitation, which, while relevant, detracts from the main focus on model control. Many large-scale generation methods today handle such limitations through increased data rather than relying solely on control refinement, making the extensive discussion on data constraints feel somewhat misaligned with the core topic.

Furthermore, the logical flow among the assumptions regarding the limitations of continuous estimators and the efficacy of diffusion models is not always clear.

Another concern is the limited experimental scope, which is restricted to 2-bar piano accompaniment generation with objective analysis only. This narrow experimental setup raises questions about whether the theoretical claims fully align with the practical findings, leaving some aspects of the methodology’s broader applicability and impact unexplored.

**Questions:**

Firstly, the paper includes several strong assertions accompanied by formal proofs. These proofs often attempt to validate assumptions that, in themselves, seem quite strong. As the strength of an assumption increases, so does the complexity and breadth of considerations required, which inherently challenges the validity of the proof. Could the authors elaborate on the rationale behind adopting such strong assumptions, and clarify whether alternative, less restrictive assumptions might yield similarly meaningful insights?

Additionally, the study is limited to generating 2-bar piano accompaniments. Why was this specific scope chosen, and could a broader scope enhance the analysis? It would be helpful to see more extensive evaluation or discussion regarding this choice, as it raises questions about the generalisability of the findings. Given that the intended audience for this paper is likely interested in musical outcomes as well as quantitative results, has there been any consideration of presenting more in-depth musical analyses or subjective evaluations alongside the objective metrics?

---

> ### Author Response · Authors · 2024-11-26
> **Response to Reviewer XuiB**
>
> We thank the reviewer for the valuable suggestions. We have modified our manuscript and have the following comments:
>
> **Reply to the concerns in “Summary”**
>
> 1. About “data scarcity as a limitation”
>
> We appreciate the reviewer’s observation regarding the focus on data limitations in the introduction. However, we respectfully clarify that the discussion of high-quality data scarcity is directly relevant to a central topic of our paper: enhancing control within symbolic music generation. Our point is that the unique challenges posed by data scarcity in symbolic music generation are inherently tied to the need for precise control mechanisms, as explained below:
>
> High-quality music data are inherently more scarce than some other modalities. Unlike domains such as image or text generation with far more abundant data, in the music domain, assessing and creating high-quality symbolic music requires significant expertise and effort. For instance, while anyone with a camera can create countless photographs, crafting new, high-quality music compositions demands substantial musical experience and creativity. This disparity underscores why symbolic music datasets are inherently limited in scale and quality. Thus, even if we can scale up the data, the amount of available data will always be relatively much more scarce than other modalities such as text and image.
>
> Furthermore, we described in our introduction that symbolic music generation requires an exceptional level of precision. Unlike images or text, where minor errors might be inconspicuous or tolerable, in tonal music, even a single incorrect note can be jarring and immediately noticeable, even to listeners with minimal musical training. This strict demand for accuracy amplifies the impact of limited training data, as errors such as "wrong notes" are far more significant in symbolic music.
>
> Empirical evidence demonstrates the limitations of scaling up data. We added Table 2 in our manuscript, showing that a model without sampling control unavoidable generates wrong notes. This aligns with the theoretical insights presented in Section 3.1, Proposition 1, where we mathematically demonstrate that with finite data, the occurrence of wrong notes is unavoidable, and the decay rate of this error probability with respect to dataset size is slow ($O(n^{-1/KL})$). Thus, relying on scaling up data alone is inefficient and impractical in symbolic music generation, given the inherent scarcity of high-quality data and the slow decay rate of errors.
>
> In contrast, our control mechanisms offer a more effective solution. Our method addresses this issue by introducing a fine-grained control mechanism that can precisely eliminate wrong notes, as demonstrated in our experiments. By guiding the generation process, we achieve harmonic consistency without needing to rely on a significant expansion of training data, which would be both resource-intensive and inefficient for this domain.
>
> Thus, the discussion on data scarcity is indeed central to our argument. It provides essential context for why control mechanisms are necessary and more effective in this domain compared to other domains. We hope this clarification strengthens the alignment between the discussion of data limitations and the primary contribution of our work.
>
> 2. About “using diffusion model”
>
> We admit that there are methods other than diffusion models for symbolic music generation. However, diffusion model is our main focus because our sampling control is plugged in the diffusion denoising process. Moreover, existing literature [1,2] has discussed and demonstrated the power of diffusion models in the related domain.
>
> 3. About “approach centers on adding fine-grained controls during training”
>
> We appreciate the comment and believe this is a misunderstanding of our contribution. We argue that our approach is not simply adding fine-grained controls during training. There are two subsections in our Section 4, and fine-grained control in training is only one subsection. The other subsection is Fine-grained control in sampling process, which is the part we put more emphasize on. Such a sampling control is different from all established control techniques because all of them are in the training process and only obtains “soft control”.
>
> **References**
>
> [1] Ziyu Wang, Lejun Min, and Gus Xia. Whole-song hierarchical generation of symbolic music using cascaded diffusion models. The Twelfth International Conference on Learning Representations,2024.
>
> [2] Yujia Huang, Adishree Ghatare, Yuanzhe Liu, Ziniu Hu, Qinsheng Zhang, Chandramouli S Sastry, Siddharth Gururani, Sageev Oore, and Yisong Yue. Symbolic music generation with non-differentiable rule guided diffusion. arXiv preprint arXiv:2402.14285, 2024.

---

> ### Author Response · Authors · 2024-11-26
>
> 4. About “that noise correction can consistently limit off-key notes, that fine-grained control will not disrupt local patterns, and that early stopping time can be practically achieved.”
>
> This appears to a misunderstanding. We would like to clarify that these are not assumptions for deriving the proposition, but instead the takaways indicated by the proposition. In our proposition 2, we have theoretically proved that the sampling control does not disrupt the distribution, and it consistently limit out-of-key notes by the construction (all out-of-key notes would be removed). This is also shown in the quantitative results and the examples on our demo page. Chord similarities increase after applying the sampling control, implying that harmonic patterns are not disrupted. As for early stopping time, this is commonly used and already well-stated in literature.[3,4] In fact, all practical diffusion models are a discretization of a SDE, and thus all diffusion methods are indeed using early stopping time because in practice one cannot sample for infinitely many time steps.
>
> 5. About “the study’s experiments are limited to 2-bar piano accompaniment generation”
>
> First, splitting the samples into 2-measure slices is only applied when calculating the metrics. In the generation process, our model is based on 4-measure slices. Even that, we can actually apply the inpainting methods of e.g., WholeSongGen[1] to create longer music out of 4-bar sections, but this is not the focus of this paper.
>
> **Reply to the concerns in “questions”**
>
> 1. We have revised our theoretical part by revising the mathematical modeling, formulation and removing as much assumptions as possible. The overall key intuition implied by the mathematical analysis is to characterize the challenge of symbolic music generation due to a combination of scarcity of data, high dimensionality and requirement for pixel-wise precision.
>
> 2. Scope of 4-bars: Again, we remark that we can actually apply the inpainting methods of e.g., WholeSongGen to create longer music out of 4-bar sections, but this is not the focus of this paper. We appreciate your suggestion of adding more in-depth musical analysis and subjective evaluations. While having a comprehensive comparison study of subjective evaluations is challenging, we plan to explore ways of adding sensible subjective evaluations or other non-quantitative alternative analysis for music.
>
> **References**
>
> [1] Ziyu Wang, Lejun Min, and Gus Xia. Whole-song hierarchical generation of symbolic music using cascaded diffusion models. The Twelfth International Conference on Learning Representations,2024.
>
> [2] Yujia Huang, Adishree Ghatare, Yuanzhe Liu, Ziniu Hu, Qinsheng Zhang, Chandramouli S Sastry, Siddharth Gururani, Sageev Oore, and Yisong Yue. Symbolic music generation with non-differentiable rule guided diffusion. arXiv preprint arXiv:2402.14285, 2024.
>
> [3]Yang Song and Stefano Ermon. Improved techniques for training score-based generative models. Advances in neural information processing systems, 33:12438–12448, 2020.
>
> [4]Alexander Quinn Nichol and Prafulla Dhariwal. Improved denoising diffusion probabilistic models. In International conference on machine learning, 2021.

---

> > ### Author Response · Authors · 2024-11-28
> > **Subjective Evaluation Updated**
> >
> > Thank you very much for your comments and suggestions.
> >
> > We have added an initial group of subjective evaluations in Section 5.1.4 of the latest version of our manuscript. We generated samples using different methods and presented to the participants in a randomized order, and their sources are not disclosed to them. The quality of samples are assessed in the following dimensions: creativity, harmony (whether the accompaniment is in harmony with the melody), melodiousness, naturalness and richness, together with an overall assessment.The average ratings and confidence intervals for six different aspects (details described in Appendix F). The results suggested that our method outperforms the baselines in all 6 different aspects.

---

> > > ### Comment · Reviewer_XuiB · 2024-11-29
> > >
> > > Thanks. Then I will keep the score of 8 for you.

---

> > > > ### Author Response · Authors · 2024-11-29
> > > >
> > > > Thank you for your support. We truly appreciate it!

---

### Official Review · Reviewer_yCUF · 2024-11-06

**Soundness:** 3
**Presentation:** 2
**Contribution:** 3
**Rating:** 6
**Confidence:** 3

**Summary:**

The paper presents a conditional diffusion model for symbolic music generation which uses chord and rhythm information as guidance. The model generates music using the piano-roll representation which the authors state is conducive to the use of diffusion models. The authors state a few difficulties in modeling symbolic music as compared to other domains, such as the need for harmonic precision, and rhythmic regularity. To overcome these issues, the model is trained using classifier-free guidance with two settings for conditioning: one where both chords and rhythm are provided, and one where only chords is provided. Additionally, the authors incorporate the key signature during sampling to avoid out-of-key notes

**Strengths:**

The paper is well written for the most part and does a very good job of motivating the problem and contrasting the presented work with other domains that typically use diffusion models.
The authors present simple approaches to overcome some of the challenges in generating symbolic music.

**Weaknesses:**

The evaluation is a little lacking. There is no ablation comparing the same diffusion model with only chord conditioning vs chord + rhythm vs unconditional. Although it might be expected that quality will drop, obtaining evidence is important.
The paper ends rather abruptly without the authors drawing clear conclusions from their study. I would recommend cutting short the diffusion model details in page 3 and dedicating some space for this.
While the ideas are interesting, they are very specific to the music domain and it is not clear how insights from this paper might benefit other fields of ML.

**Questions:**

See weaknesses section.

---

> ### Author Response · Authors · 2024-11-26
> **Response to Reviewer yCUF**
>
> We thank the reviewer for the suggestions. We have made the following modifications to our manuscript:
>
> 1. We have added a section 5.1.5 of ablation studies. We run 3 additional experiments on the same accompaniment generation task to analyze the impact of the fine-grained conditioning during training and the fine-grained control in sampling. Tur first experiment involves the same model trained with fine-grained conditioning but only removes the out-of-key notes after the last sampling step; the second also incorporates fine-grained conditioning for training but without any control during sampling; the third is an unconditional model without any conditioning or control in both the training and sampling process. All experiments use the same model architecture and random seeds as the one with full control for comparability. We evaluate the frequency of out-of-key notes and assess the overall performance using the same quantitative metrics as before. We find that conditioning in training and control in sampling both contribute to reducing out-of-key notes and improving overall performance.
> Moreover, in Figure 6 of our manuscript, we provide two examples showing that our sampling control can not only eliminate incorrect notes, but also guide the model to replace them with more harmonic ones.
>
> 2. We refined the theoretical part to be significantly shorter, and added a conclusion section to summarize our work as well as discuss the limitations of our method.
>
> 3. For other fields of ML, in our future work we plan to analyze a more general framework discussing the joint effect of conditioning and sampling guidance in diffusion models. We will discuss the application in more domains such as AI safety.

---

> > ### Author Response · Authors · 2024-11-28
> >
> > Additionally, for better evaluation, we added an initial group of **subjective evaluations** in Section 5.1.4. We presented average ratings and confidence intervals for six different aspects (details described in Appendix F). The results suggested that our method outperforms the baselines in all 6 different aspects.

---

### Author Response · Authors · 2024-11-28
**General response about paper revision**

Dear reviewers,

We would like to provide a brief outline of the modifications in the revised manuscript. Some modifications were made in the first revision (revised on November 25), after the first round of reviews became available. Some other experiments and evaluations were further added in the current revision (Nov 27, 2024), after the more recent discussions. The main changes are summarized as following:

**Changes made in the November 25 version:**

1. We refined the theoretical part to be significantly shorter by revising the mathematical modeling, formulation and removing as much assumptions as possible.

2. We added an **ablation study** section (which is now Section 5.1.5 in the latest version), where we compared our full method (training condition + sampling control) with 1) training condition + post-sample editing, 2) training condition but no sampling control, 3) unconditional model and no sampling control. Our full method outperforms all of them, showing that conditioning in training and control in sampling both contribute to the improvement of performance.

3. We added a conclusion section to summarize our work as well as discuss the limitations of our method.

4. Since we conducted additional experiments that requires retraining, we retrained our models and changed the number of epochs from 10 to 20. This results in an improvement on the evaluation metrics, and we have updated them in our table 1.

**Changes made in the November 27 version:**

1. We added an initial group of **subjective evaluations** in Section 5.1.4. We presented average ratings and confidence intervals for six different aspects (details described in Appendix F). The results suggested that our method outperforms the baselines in all 6 different aspects.

2. We have revised the descriptions and logic of our work to emphasize "control" rather than "correctness." Out-of-key notes are not inherently "wrong"; instead, the issue of degraded quality in generating out-of-key notes serve as an illustrative example of the challenges that current models face in accurately capturing distributional information in symbolic music data. Furthermore, the external sampling control is not limited to correcting "wrong notes" but is designed to provide a more efficient and reliable framework for aligning the model's output more precisely with the user's intent.

3. In response to reviewer aY42, we added some experiments in Appendix H about the effect of classifier-free guidance weights for rhythmic condition. The results shows a trade-off between sample quality and adherence to the condition, which is consistent with the findings in existing literature.

4. In response to reviewer aY42, we have updated some experiments in Appendix E.3 about symbolic music generation tasks (where both melody and accompaniment are generated) conditioning solely on chord progression. The results indicate that our FTG method still outperforms baselines on this task.

**Additional changes on the demo page:**

1. In Section 1 of our demo page, we added examples demonstrating the application of our method in creating stylized music (including Chinese style and Dorian mode) by conditioning on chord and rhythm and leveraging scale-based control. Such styles and modes do not exist in the training set.

2. In Section 2 of our demo page, we showcase the effectiveness of sampling control by providing some examples that compare the results with/without sampling control.

---

### Meta-Review · Area_Chair_NSoq · 2024-12-20

**Metareview:**

The paper presents a method for adding chord and rhythm constraints in symbolic diffusion, as a way to avoid false notes and provide a more fine-grained control over the generation results. As part of the problem statement, the authors refer to new statistical bounds on conditional probability matching in DDPM with CFG. These broad motivations seem reasonable as motivation but are not translated to actual quantitative results or related specifically to the piano-roll diffusion problem. The main contribution of the paper is suggesting novel loss functions for model training and a method for sampling correction during inference, so as to better match the given constraints.
An interactive demo is available on huggingface, capable of producing relatively short generation examples. Human evaluation was provided as part of the rebuttal process.

**Additional Comments On Reviewer Discussion:**

The paper received a large variety of rankings, with two rejects, one strong accept and one borderline above threshold ranking. Moreover, reviewer aY42 raised concerns about integrity of some of the new results during rebuttal, which they expressed as an ethics issue. Another concern in my consideration of the strong accept review is that it seems to have used chatGPT or similar to write the paper summary, though the questions appear to be genuine.  The confidence of the reviewers was also widely varied, which created an overall situation that put more weight on the meta-review.
My impression is that the difficulty of the paper is largely in presenting a concise problem statement and a focused discussion of the main contribution. The motivation section based on sharp statistical theoretical analysis might have been unnecessary as it complicated the presentation without showing direct relevance to the proposed solution. Moreover, the paper itself was significantly modified during the rebuttal, which suggests that further refinement of the presentation might be required.  As to the ethics concern, since the onus of proof that results might have been reported selectively is the responsibiliy of the reviewer, I could not take this concern into consideration.
Overall, while the topic is relevant and the paper has some valuable original algorithmic and theoretical contributions, concerns about limited experimental evaluation and unclear presentation remain. Accordingly, the paper does not seem to meet the conference’s standards at this time and I recommend rejection, but wouldn't mind bumping it up as a borderline case.

---

### Decision · Program_Chairs · 2025-01-22

Reject